# Spaced Scheduling for
# Large Language Model Training

**Amine El Hattami**                                    *amine.elhattami@servicenow.com*
*ServiceNow Research, Mila, Polytechnique Montréal*

**Nicolas Chappados**
*ServiceNow Research*

**Christoper Pal**
*ServiceNow Research, Mila, Polytechnique Montréal, Canada CIFAR AI Chair*

**Reviewed on OpenReview:** *https://openreview.net/forum?id=pOKTYl2B9T*

## Abstract

Recent breakthroughs in deep learning have accelerated progress toward increasingly capable large language models (LLMs), even sparking discussions about the path to Artificial General Intelligence (AGI). Yet, current LLM training pipelines continue to depend on heuristics and human-driven empirical analysis to curate data. In practice, more sophisticated data selection methods often incur high costs, exhibit limited adaptability, or do not consistently surpass simple random baselines across various models and datasets. In this work, we propose *Spaced Scheduled Training* (SST), a novel adaptive data selection strategy that prioritizes training examples based solely on per-example perplexity computed from the model's own evolving parameters. By obviating the need for external reference models, SST customizes data selection to the model's unique characteristics, including its pre-training data composition, and eliminates biases commonly introduced by these external models. Extensive experiments on seven LLMs (0.5B to 32B parameters) in the instruction-finetuning (IFT) setting show that SST consistently outperforms representative state-of-the-art selection approaches like DEITA and INSTAG on the Open LLM Leaderboard. For instance, with Qwen2.5-32B and a 30k examples data budget, SST achieved a 42.75% Open LLM Leaderboard score, exceeding a leading data-selection baseline (38.56%) and the full-100k dataset baseline (39.58%). We further present a theoretical framework to assess computational overhead of model-based selection methods, showing that SST remains efficient in practical scenarios, and propose strategies to mitigate the overhead in worst-case scenarios. Our findings underscore the potential of model-informed dynamic data selection, offering an efficient, adaptable, and cost-effective approach. We release our training code, trained models, and data mixes in our public repository[1].

## 1 Introduction

Recent advances in large language models (LLMs) have transformed natural language processing, enabling breakthroughs in applications ranging from artificial agents to scientific discovery. While scaling the model size and the training data has driven much of this progress, *data quality* is increasingly recognized as a bottleneck, particularly when hardware and computational budgets constrain further scaling. Data selection methods can be broadly categorized into two approaches: *static* and *dynamic*. Static approaches pre-select data offline using heuristics or external "oracle" models. These methods often incur substantial computational cost, either from training specialized evaluation models (Liu et al., 2024; Lu et al., 2023) or from using

---

[1]https://github.com/Am1n3e/sst

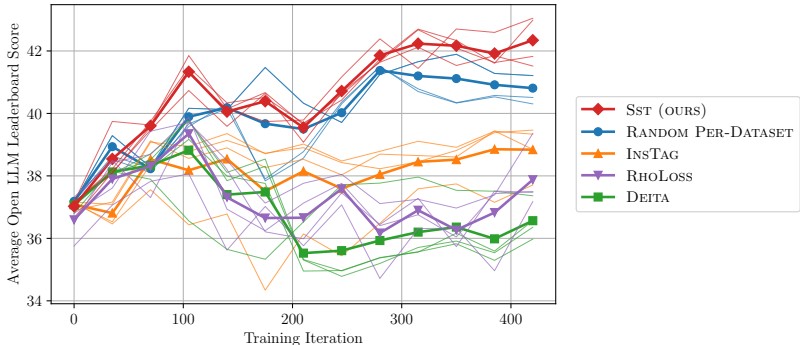

Figure 1: Performance comparison on Qwen2.5-32B (30k instruction budget for 2 epochs). Sᴄ significantly outperforms a strong random baseline, competitive ChatGPT-based methods (Iɴsᴛᴀɢ, Dᴇɪᴛᴀ) and another adaptive method (Rʜᴏ-Lᴏss). Despite Qwen2.5-32B often being considered less sensitive to sampling approaches, Sᴄ achieves higher Open LLM leaderboard[2] scores throughout training and displays lower variance across four random seeds. The late-stage uptrend suggests further training may extend Sᴄ's lead. All methods draw 30k examples from the same 100k-instruction pool.

costly commercial LLMs for data scoring (Chen et al., 2024). Moreover, oracles can introduce biases in the evaluation, such as the documented verbosity bias in ChatGPT (Saito et al., 2023). This bias is particularly problematic when the same oracle is used for benchmarking, as is the case with MT-bench (Zheng et al., 2023). While these methods can yield reasonable performance, they lack adaptability: the dataset remains fixed regardless of the trained model's characteristics (e.g., size, or pre-training data composition). Dynamic approaches, in contrast, integrate data selection *during* training (Mindermann et al., 2022; Jiang et al., 2019; Loshchilov & Hutter, 2015), which can, in principle, adapt the curation to the model's evolving state. Yet many of these techniques are expensive to run at scale or fail to provide consistent performance. For instance, RHO-LOSS (Mindermann et al., 2022) requires training a proxy model, performing a forward pass on the entire training dataset using this proxy model, and conducting additional forward passes for batch selection. Further, studies have shown that many dynamic methods fail to consistently outperform simple random selection baselines (Kaddour et al., 2023; Liu et al., 2024). As a result, recent works like Tülu 3 (Lambert et al., 2024) still rely on heuristics and large-scale empirical analysis to refine data. Recent research highlights additional challenges in data selection, especially when dealing with diverse data sources. Determining the optimal mixing ratio is difficult and depends on the specific model being trained. Further, relying on external oracle models can introduce unintended biases, as they may overemphasize data that aligns well with their own output. For instance, Dᴇɪᴛᴀ (Liu et al., 2024) sampled a 6k-example subset from a 300k-example pool comprising three sources (ShareGPT (Chiang et al., 2023), UltraChat (Cui et al., 2024), and WizardLM (Xu et al., 2023)); however, this subset exclusively contained examples from ShareGPT. Recent LLM pre-training work, such as Llama 3 (Dubey et al., 2024) suggests that adjusting mixing ratios multiple times during training is beneficial. This observation highlights the importance of dynamic approaches and reinforces the need for more adaptive methods.

To address these challenges, we propose a novel adaptive and efficient data selection strategy called Spaced Scheduled Training (Sᴄ) that dynamically adjusts the training dataset based on a model's evolving learning state. Unlike approaches that rely on external scoring models, Sᴄ prioritizes training examples based on per-example perplexity, a computationally efficient and reliable proxy for example difficulty. Our work builds on prior work in static selection (Marion et al., 2023) but incorporates dynamic scheduling to continuously adjust the dataset mix throughout training. Sᴄ also differs from existing work in that it: (i) eliminates the need for costly external oracle models; (ii) tailors selection to the target model's unique characteristics (e.g., size and pre-training data composition), avoiding biases from external scoring models; and (iii) adapts the dataset mix continuously through its "spaced scheduling" mechanism, allowing different models to emphasize data that is most beneficial *at each stage* of training. We evaluate Sᴄ across **seven LLMs** with sizes ranging from **0.5B** to **32B** parameters from **four distinct model families**, including Llama 3.1 (Grattafiori et al., 2024a), Llama 3.2 (Dubey et al., 2024), Gemma 2 (Team et al., 2024), and Qwen 2.5 (Yang et al., 2024). Our

empirical analysis, using the recent Open LLM Leaderboard (Fourrier et al., 2024) demonstrates that SST delivers consistent performance across architectures and model sizes. As illustrated throughout training in Figure 1, SST *outperforming all selection baselines*, including those relying on ChatGPT-based selection (Liu et al., 2024; Lu et al., 2023). Furthermore, we introduce a theoretical framework, grounded in LLM scaling laws (Kaplan et al., 2020), to quantify the scaling of overhead for data selection methods, including SST. This framework allows for a more principled comparison of different approaches. Through this framework, we show how inference-optimized backends allow SST to maintain *low computational overhead, enabling large-scale use*. We summarize our key contributions as follows:

- **Novel Dynamic Data Selection Algorithm (Sst)**: We introduce Spaced Scheduled Training (SST), a novel approach that relies on per-example perplexity to prioritize training examples and dynamically adjust the data mix throughout training (§ 4).
- **Comprehensive Empirical Evaluation**: We demonstrate SST's effectiveness in the IFT setting on seven models (0.5B to 32B parameters). SST outperforms all baselines on five out of seven models, including the full 100k dataset baseline. On the remaining two models, SST achieves average Open LLM Leaderboard scores within 1.14 absolute points of those from the full 100k baseline (§ 5).
- **Scalability and Theoretical Overhead Analysis**: We introduce a framework, based on LLM scaling laws research to analyze the overhead scaling of data selection methods, including SST. We demonstrate how to implement SST efficiently for large-scale training (§ 4.2).
- **Practical Insights for LLM Data Selection**: We analyze perplexity-based signals to provide foundational insights and guidelines for future work in data selection. Our findings show that perplexity sampling is influenced by factors such as subset size and pre-training data composition. Furthermore, it is beneficial to delay perplexity sampling until training stabilizes, and a dynamic policy for selecting data leads to improved results (§ 3).

## 2 Related Work

Data selection plays a crucial role in training large language models (LLMs). It aims to prioritize informative examples or remove non-useful or noisy data that might degrade performance. Data selection approaches fall into two main categories: static and dynamic methods. This section reviews key research in data selection, highlighting their strengths and limitations, and motivating our proposed SST approach.

**Static methods** pre-select the training data offline, before training begins, independent of the specific model being trained. Marion et al. (2023) proposed a static pruning method relying on perplexity (PPL) scores computed with an external reference model. They demonstrated that PPL is more effective than more complex metrics, such as Error L2-Norm (EL2N). By pruning examples with low PPL, they achieved similar performance using only 30 % of the data. Sorscher et al. (2022) explored pruning in computer vision, using proximity to the decision boundary as a difficulty measure in a teacher-student perceptron setup. They showed that the examples chosen for pruning depend on the initial dataset size and identified the conditions for aggressive pruning. They emphasized that the success of pruning methods relies on the quality of the pruning metric, noting that most metrics they tested are costly to compute, making them impractical for large-scale use. Sachdeva et al. (2024) explored pruning for pre-training T5 models (Raffel et al., 2020), introducing two scoring methods: (1) DENSITY, which estimates whether similar examples have been sampled, and tries to maximize coverage; (2) ASK-LLM evaluates example quality by prompting an instruction-tuned LLM (FLAN-T5; see Chung et al. (2022)) to predict whether it contains informative signals (a yes/no question). The study demonstrated that DENSITY performs comparably to using the full dataset. In contrast, ASK-LLM outperforms full-data training by rejecting 90 % of the data and converging 70 % faster. Recent instruction fine-tuning (IFT) work leverages state-of-the-art commercial models to score and select examples (Chen et al., 2024; Liu et al., 2024; Zhao et al., 2023; Xu et al., 2023). ALPAGASUS Chen et al. (2024) uses ChatGPT with a handcrafted prompt to predict scores from 0 to 5, pruning the examples below a certain threshold. DEITA (Liu et al., 2024), building on Xu et al. (2023), adds diversity as a selection criterion and proposes a two-level scoring system: Evolve Complexity $c$ and Evolve Quality $q$, to compute a single score $s = c \times q$. Initially, ChatGPT is used for scoring, but Liu et al. (2024) later trains a model to replicate ChatGPT's scoring, substantially reducing the cost. DEITA shows that a mere 10,000 examples can outperform models trained on ten times as many. Similarly, INSTAG (Lu et al., 2023) uses intention

tags as metrics for instruction diversity and complexity. It utilizes ChatGPT to assign one or more tags to each instruction example. To ensure high-quality tags, the tags are normalized using frequency filtering and aggregation (e.g., semantic aggregation). INSTAG prioritizes complex queries with the highest number of tags, while maintaining diversity by selecting examples that expand tag coverage. To reduce the computational cost of tag assignment, the authors created INSTAGGER, a distilled LLM that mimics ChatGPT annotation capabilities. We compare our approach with DEITA and INSTAG in our experimental work.

Model-based static approaches currently produce the best results. However, despite reducing manual effort associated with empirical approaches which involve extensive human intervention, they often incur high computational costs. When training specialized evaluation models, the cost of these models can be offset by repeated use, but this advantage diminishes in practice. For instance, DEITA (Liu et al., 2024) requires training two specialized scoring models that have a maximum sequence length of 2,048; handling longer sequences requires re-training a new evaluation model. API-based methods, like ALPAGASUS (Chen et al., 2024), are susceptible to prompt design issues and biases such as the verbosity bias inherent in GPT models (Saito et al., 2023), leading to imbalanced data selection, as illustrated by the over-representation of a single source dataset in DEITA's curated 6k examples subset containing 100% of the examples from the ShareGPT dataset. These issues are particularly problematic when the same models are used for both data curation and benchmarking (e.g., MT-bench; Zheng et al. (2023)) as performance can be artificially inflated. Beyond computational overhead, static dataset selection methods suffer from a fundamental limitation: they fail to account for the model's unique pre-training data composition. For instance, if a model is pre-trained on 70% code-related data, an effective selection strategy would need to down-weight code-related tasks. This also applies to the data complexity categorization, where an example requiring a 10-step chain-of-thought reasoning process may be considered easy for a 70B model but intractable for a 1B model.

**Dynamic methods** integrate data refinement directly into the training process. For instance, online batch selection aims to optimize training by selecting examples for each batch, often by scoring and ranking a large batch to select the top-$k$ examples (Loshchilov & Hutter, 2015). Jiang et al. (2019) proposed Selective-Backprop, which prioritizes examples with high loss. However, Mindermann et al. (2022) challenged this approach, arguing that high-loss examples can be noisy or mislabeled. They introduced the Reducible Holdout loss selection (RHO-LOSS) (Mindermann et al., 2022) which reduces the impact of noisy data by weighting down their losses. RHO-LOSS uses a proxy model trained on a holdout set to select examples that minimize the holdout loss by approximating the reducible holdout loss objective. While RHO-LOSS improved accuracy and training speed, it incurs significant overhead due to the cost of training the proxy model, a forward pass on the entire training data using the proxy model, and the additional forward passes for the batch selection. Dynamic methods attempt to overcome the rigidity of static approaches by selecting data during training. However, despite their adaptability, these methods often fall short of state-of-the-art static model-based approaches. For instance, Kaddour et al. (2023), found that RHO-LOSS (Mindermann et al., 2022) fails to outperform simple random selection baselines consistently. Moreover, dynamic methods also suffer from algorithmic complexity that prohibits their use at scale. The aforementioned overhead of RHO-LOSS, for example, makes it impractical at scale.

In both categories, when the initial data pool contains multiple sources or categories, determining the optimal mix ratio creates an additional complexity for any data selection method. Finding the *optimal* mixture remains underexplored and currently relies heavily on heuristics and extensive empirical analysis to select the mixing ratio. Further, recent work in Llama models (Dubey et al., 2024) shows that adjusting the mix ratio multiple times during pre-training can be beneficial, as it enables the model to better adapt to its evolving needs. The shortcomings discussed above underscore the necessity for an adaptive data selection strategy that dynamically tailors the training data to the evolving needs of the model while preserving computational efficiency at scale, forming the foundation for our proposed method, Spaced Scheduled Training (SST), detailed in §4. SST leverages the target model itself to guide the data selection process. By eliminating the need for costly external models, SST adapts both data categorization and selection criteria to the characteristics of the trained model. Our approach uses per-example perplexity as a computationally efficient and reliable proxy for example difficulty, a premise we rigorously examine in §3 and Appendix D, with its overhead further analyzed in §4.2. By extending the static selection method of Marion et al. (2023) with a dynamic mechanism that continuously adjusts the dataset mix throughout training, SST overcomes

the limitations of previous methods and better aligns the training data with the model's evolving state. These improvements enable SST to outperform the best existing methods (INSTAG and DEITA) and even baselines utilizing significantly more training data, as shown in §5.

## 3 Preliminary Analysis

This section presents the key findings of our preliminary analysis of perplexity-based data selection in the IFT setting. These insights motivate and lay the groundwork for our proposed adaptive method in §4.

We analyze perplexity-based data selection within the Instruction Fine-Tuning (IFT) setting, extending the approach of Marion et al. (2023) with several key distinctions. **IFT Setting**: We investigate perplexity-based data selection in the context of IFT rather than pre-training. **Target Model as Reference**: Unlike approaches using external models, we use the target model itself to guide the selection of its training data. **Broad Analysis Scope**: We evaluated models ranging from 0.5B to 32B parameters across state-of-the-art architectures (Llama3.1 (Grattafiori et al., 2024a), Llama3.2 (Grattafiori et al., 2024a), Qwen2.5 (Yang et al., 2024), and Gemma2 (Team et al., 2024)), offering a significantly broader evaluation than the two models used in Marion et al. (2023), totaling 248 training runs across different sampling settings and seeds. We compare static perplexity-based sampling and random selection (using 10%, 30%, and 50% of a pool of 100,000 examples) against a baseline trained on the full dataset, with performance measured using the Open LLM Leaderboard benchmarks. We sample from a dataset collection $\mathcal{D} = \{D_i\}$ where $D_i$ is an instruction dataset. The collection consists of 15 datasets from the TÜLU 3 mixture (Lambert et al., 2024). We compute the per-example perplexity values $\text{PPL}(e)$ on the target tokens only, to align with the IFT setting as follows:

$$\text{PPL}(e) = \exp\left(\frac{1}{|e|}\sum_{t_j \in e}\text{NLL}(t_j)\right) = \exp(L_e), \tag{1}$$

where $\text{NLL}(t_j)$ denotes the negative log likelihood of output token $t_j$, and $L_e$ is the target loss of the example $e$. Detailed experimental settings and further analysis are provided in Appendix D. The key findings of our analysis are threefold. First, **heuristic-based curation is not sufficient**: as illustrated in Figure 2, simple random sampling can sometimes outperform training on all 100,000 examples, suggesting that even a carefully curated data mix (Lambert et al., 2024) contains redundant or less beneficial examples.

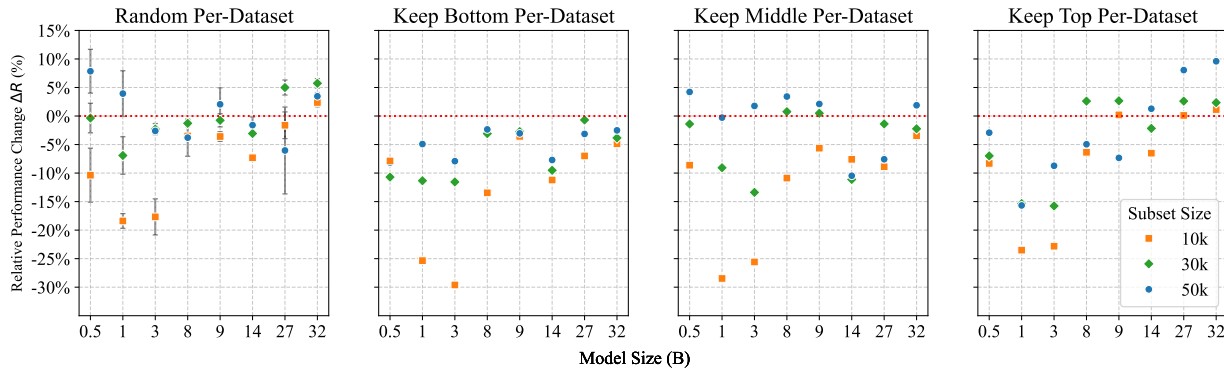

Figure 2: Performance of static per-dataset perplexity sampling: Middle-segment tends to improve smaller models (<8B), while top-segment benefits larger models (e.g., Qwen2.5-32B). Only certain static perplexity-based configurations surpass the baselines, motivating the adaptive approach we introduce later. Relative performance change $\Delta R$ is calculated as $\Delta R = (S_{\text{method}} - S_{\text{baseline}})/S_{\text{baseline}}$ (Equation 7), where $S_{\text{method}}$ and $S_{\text{baseline}}$ are the average Open LLM Leaderboard score of the method and 100% baseline. Points below the red dashed line indicate performance drops compared to baseline. Error bars represent standard error over two seeds. Detailed experimental setup is provided in Appendix D.

Second, **the best-performing perplexity sampling criteria vary with model size and subset size:** smaller models ($< 8$B) tend to benefit from "middle" perplexity ranges, while larger models ($> 14$B) often

gain more from high-perplexity examples (Figure 2). Further, this criterion varies with subset size. Figure 2 shows that using examples from the bottom segment is not beneficial across models. When selecting a large subset (e.g., 50% of the data) for a model that typically benefits from mid-perplexity examples, choosing these examples predominantly from the top perplexity segment can be more advantageous than attempting to draw such a large fraction from only the middle segment. The latter approach could incorporate too many low-perplexity (easy) examples from the broadened middle segment, thereby negatively impacting performance. This trade-off also applies when selecting a smaller subset, such as 10% of the examples. Finally, **aligning data complexity with training progress proves beneficial**. As shown in Figure 3, different perplexity segments are most useful at various training stages, where the models benefit from beginning with easier examples and then transitioning to more challenging ones, which mirrors curriculum learning principles (Bengio et al., 2009). Further, we found it beneficial to start training with randomly selected data per dataset until a set training iteration threshold is reached. After this point, we compute the perplexity values once and initiate perplexity sampling. This delay ensures that perplexity sampling starts after the model's data shift adaptation in early training, ensuring that the sampling uses more meaningful perplexity values. As shown in Figure 4, the delay varies across models and decreases as model size increases, and delaying beyond this threshold harms performance due to changes in the perplexity distribution, as we describe in more detail in Appendix D.3.

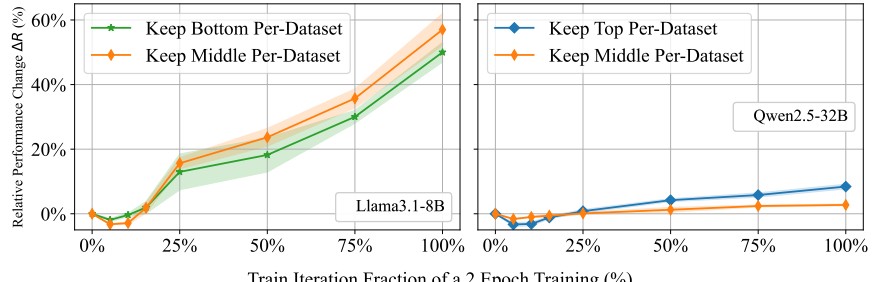

Figure 3: Performance comparison of static perplexity segments during training for Llama3.1-8B (left) and Qwen2.5-32B (right). Models benefit from starting with easier examples (lower perplexity) before transitioning to harder ones (higher perplexity), motivating dynamic selection strategies akin to curriculum learning (Bengio et al., 2009). The figure displays the best-performing and second-best performing static configurations identified for each model (Appendix D). Performance is measured as relative change $\Delta R$ (Equation 7) of Open LLM Leaderboard mean scores compared to the base pre-trained model. Standard error over two seeds shown.

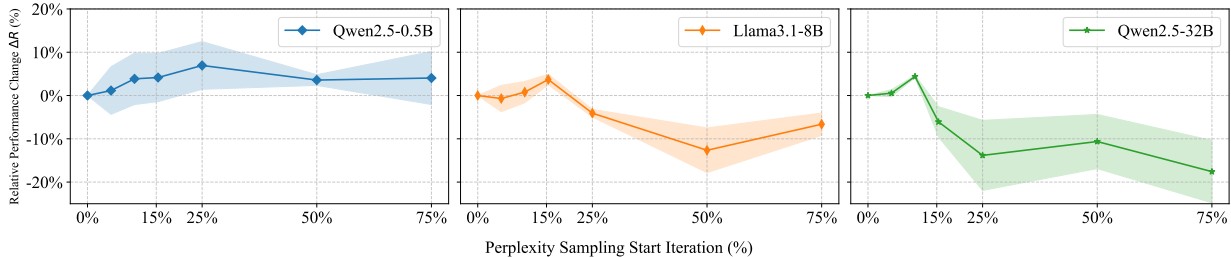

Figure 4: Effect of delaying static perplexity-based data selection. Delaying perplexity sampling improves performance up to a threshold that varies by model and generally decreases with model size. Smaller models (e.g., Qwen2.5-0.5B) benefit from a longer delay (25% of training), while larger models (e.g., Qwen2.5-32B) degrade as early as 10%. $\Delta R$ (Equation 7) is the relative performance change percentage compared to using no delay (using the pre-trained model for data selection). Results show mean scores and standard error across two runs with different random seeds.

Together, these results motivate our dynamic data selection strategy that delays perplexity sampling until the model starts producing meaningful and reliable perplexity values and refines which perplexity segments

are most beneficial as training progresses, rather than relying on a single static approach. Taken together, these *preliminary* findings highlight the limitations of static perplexity-based selection methods, emphasizing that no single static strategy consistently benefits all models across training stages. Smaller models benefit from different perplexity segments compared to larger models, and the optimal choice varies dynamically during training. Furthermore, delaying perplexity-based selection until models stabilize yields significant performance improvements, underscoring the need for an adaptive approach. Motivated by these observations, we propose Spaced Scheduled Training (SST) (§4) to mitigate the limitations of static perplexity-based selection methods.

## 4    Spaced Scheduled Training

We propose Spaced Scheduled Training (SST), an efficient and adaptive method for selecting and scheduling training examples based on their perplexity. SST, as shown in Figure 5, builds on insights presented in §3, where we demonstrate that static selection strategies often fail to deliver consistent improvements. In contrast, SST provides a dynamic approach to perplexity-based data selection, tailoring this selection to the characteristics of the target model (i.e., the model being trained), such as its size and its unique— and often obscure—pre-training configuration. SST utilizes dataset-aware selection, unlike methods such as DEITA (Liu et al., 2024) or INSTAG (Lu et al., 2023), which disregard implicit dataset categorizations during the selection process. By focusing on examples most likely to improve the model's learning, SST enhances performance while optimizing resource utilization. It consistently outperforms static and heuristic-based methods, offering a scalable and practical solution for efficient data selection. Moreover, SST avoids reliance on external reference models that are costly or have restrictive data licenses (e.g., ChatGPT), additional scoring models that become outdated (Lu et al., 2023; Liu et al., 2024), or complex selection techniques that are impractical at scale (Mindermann et al., 2022; Liu et al., 2024), making it easily adaptable to existing training pipelines.

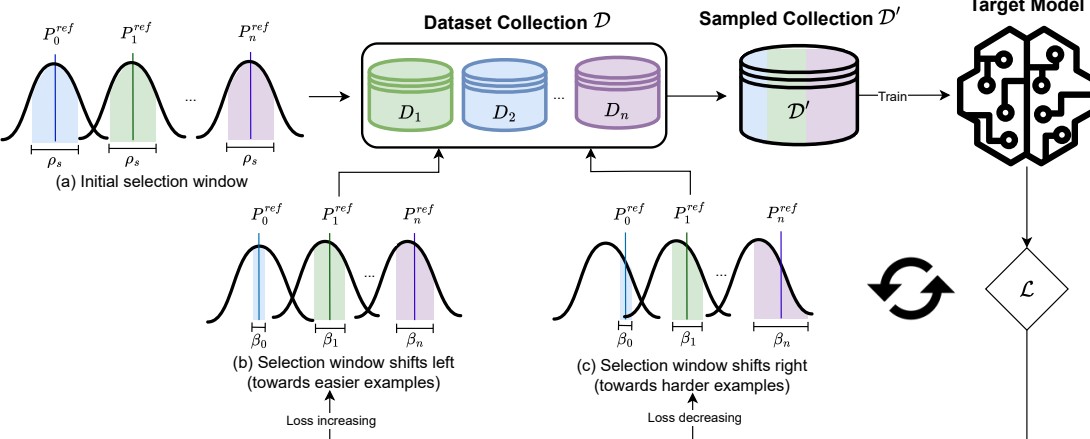

Figure 5: A visualization of the adaptive phase of our SST algorithm, using a dataset collection $\mathcal{D} = \{D_d\}$, composed of $d = 1 \ldots n$ datasets. (a) SST initializes the reference perplexity percentiles $P_d^{\text{ref}}$ for each dataset $\mathcal{D}_d$ to the 50th percentile. It then selects the initial collection $\mathcal{D}'$ based on the initial dataset ratios $\beta_d$, such that the total size of $\mathcal{D}'$ is $\rho_s|\mathcal{D}|$. Then, SST trains the model on $\mathcal{D}'$ and monitors the loss curve. (b) If the loss increases significantly, the selection window shifts toward easier examples (bottom segment), suggesting that the current data configuration is too difficult for the model's current state. (c) If the loss decreases significantly, indicating that the target model can learn from more complex examples, SST shifts the selection window toward harder examples (top segment). If the loss curve is stable, SST continues training without changes. At each shift, SST adjusts the dataset ratios using $\beta_d$ (Equation 2), prioritizing datasets with higher $\text{PPL}_{\text{p50}}(d)$ values. This process continues until the end of training.

## 4.1 Method

SST dynamically selects and schedules training examples at each stage of training, using per-example perplexity as a measure of example utility. SST operates on two levels: (1) adjusting dataset ratios to prioritize datasets and (2) filtering examples to focus on the most impactful training examples within each dataset. This dual-level approach allows SST to continuously optimize the training process, enhancing convergence speed and improving generalization for the target model. The algorithm consists of two main phases: a warm-up phase and an adaptive scheduling phase, as shown in Algorithm 1.

---

**Algorithm 1** Spaced Scheduled Training (SST) (full version in Algorithm 2).

---

**Require:** $\theta$: initial model parameters; $\mathcal{D} = \{D_1, \ldots, D_d\}$: collection of $d$ datasets; $t_{\max}$: maximum training iterations; $k_w \in (0, 1]$: warm-up slope window ratio; $r_w$: maximum warm-up slope evaluation retry count. $\rho_s \in (0, 1]$: global subset ratio; $\tau \in (0, 1]$: window-shift ratio; $\varepsilon > 0$: slope threshold for loss slope evaluation;
**Ensure:** Updated model parameters $\theta$.
1: **Warm-up phase**
2: $\theta, t_w \leftarrow \text{WARMUP}(\theta, \mathcal{D}, t_{\max}, r_w, k_w)$

3: **Adaptive Scheduling Phase**
4: $k \leftarrow t_w / t_{\max}$
5: $P \leftarrow \text{COMPUTEPPL}(\theta_{t_w}, \mathcal{D})$          ▷ Compute per-example perplexities PPL($e$) for each example $e \in \mathcal{D}$
6: Compute $\beta_d$ for each dataset $D_d$ using Equation 2
7: $P_d^{\text{ref}} \leftarrow 50$ for each dataset $D_d$          ▷ Initialize window reference to the 50th percentile
8: $\mathcal{D}' \leftarrow \text{SELECTPERDATASETSUBSET}(\mathcal{D}, \rho_s, \beta_d, P_d^{\text{ref}})$
9: Initialize loss buffer $L_{\text{buf}} \leftarrow []$ (capacity $\lfloor k \cdot t_{\max} \rfloor$)
10:          ▷ Main training loop
11: **for** $t = t_w + 1$ **to** $t_{\max}$ **do**
12:     $\theta, L_B, P \leftarrow \text{TRAINSTEP}\big(\theta, \text{SAMPLEBATCH}(\mathcal{D}')\big)$      ▷ Train model on a batch, and update batch perplexities
13:     Append $L_B$ to $L_{\text{buf}}$
14:     **if** $|L_{\text{buf}}| = \lfloor k \cdot t_{\max} \rfloor$ **then**
15:         $slope \leftarrow \text{COMPUTESLOPE}(L_{\text{buf}})$      ▷ e.g., via simple linear regression
16:         Clear $L_{\text{buf}}$
17:          ▷ 1- Recompute dataset mixing ratios
18:         Update $\beta_d$ for each dataset $D_d$ using Equation 2
19:          ▷ 2- Decide if window shifts to harder or easier examples
20:         **if** $slope < -\varepsilon$ **then**
21:             $P_d^{\text{ref}} \leftarrow \min\big(P_d^{\text{ref}} \times (1 + \tau),\ 100 - \frac{\beta_d \times 100}{2}\big)$    ▷ Loss is decreasing, shift to more complex examples
22:         **else if** $slope > \varepsilon$ **then**
23:             $P_d^{\text{ref}} \leftarrow \max\big(P_d^{\text{ref}} \times (1 - \tau),\ \frac{\beta_d \times 100}{2}\big)$    ▷ Loss is increasing, shift to easier examples
24:         **else**
25:             **no change to** $P_d^{\text{ref}}$      ▷ Loss is stable, keep the current window
26:         **end if**
27:          ▷ 3- Select updated subset based on new mixing ratios and reference perplexity percentiles
28:         $\mathcal{D}' \leftarrow \text{SELECTPERDATASETSUBSET}(\mathcal{D}, \rho_s, \beta_d, P_d^{\text{ref}})$
29:     **end if**
30: **end for**
31: **return** $\theta$      ▷ Final trained model parameters

---

Table 1: Overview of SST-specific hyperparameters used during the warm-up phase. Values are based on empirical observations (§3). Generic hyperparameters (e.g., $t_{\max}$ or $\rho_s$) common to data filtering methods are omitted.

| | Description | Value |
|---|---|---|
| $k_w$ | **Warm-up slope evaluation window size** (fraction of $t_{\max}$). Used to detect early loss trend stabilization. | 0.1 |
| $r_w$ | **Warm-up maximum retry count** for slope evaluation. Safeguard to prevent an overly long warm-up phase. | 3 |
| $\varepsilon$ | **Slope evaluation threshold**. Ensures robustness to minor fluctuations. | $10^{-3}$ |
| $\tau$ | **Selection window adjustment factor**. Controls the magnitude of the selection window shift. | 0.1 |

**Warm-up phase:** The warm-up phase stabilizes training dynamics and ensures the model generates meaningful perplexity values before transitioning to adaptive scheduling. During this phase, the model trains on uniformly sampled data across all datasets, while SST monitors the training loss curve using a rolling window of size $k_w \cdot t_{max}$, where $t_{max}$ is the maximum training iterations and $k_w \in (0, 1]$ is a hyperparameter controlling the window's proportion relative to the training duration. The warm-up ends when one of the following conditions is met: (1) the loss curve stabilizes, indicated by a nearly constant slope after the initial sharp decline in loss typical of early training; (2) a maximum retry count $r_w$ is reached, which acts as a

safeguard to limit the duration of the warm-up phase. The hyperparameter $r_w$ is particularly useful for smaller models (fewer than 8B parameters), as they may not exhibit a clearly identifiable loss stabilization point. Choosing appropriate values for $k_w$ and $r_w$ is critical: $k_w$ should allow sufficient iterations to detect changes in the loss trend, while $r_w$ prevents the warm-up from taking a significant proportion of the overall training, reducing the benefits of SST. Based on empirical observations (§3), we found setting $k_w \leq 0.25$ and $r_w$ so that the warm-up phase takes no more than 30% of the total training duration to be effective. In our experimentation, we set $k_w = 0.1$ and $r_w = 3$. This phase helps SST converge more rapidly toward a better selection window by deferring the start of adaptive scheduling until the model produces reliable perplexity values, which might otherwise be unstable during early training due to data distribution shifts or chat template adaptation in IFT.

**Adaptive Scheduling Phase:** SST performs the following during its adaptive scheduling phase: (1) Compute the example perplexity values $\text{PPL}(e)$ (Equation 1) across the entire dataset collection $\mathcal{D}$ a single time. (2) Compute the dataset mix ratios $\beta_d$. For each dataset $d \in \mathcal{D}$, the ratio is given by:

$$\beta_d = \rho_s \times \frac{\text{PPL}_{\text{p50}}(d)}{\sum_{d' \in \mathcal{D}} \text{PPL}_{\text{p50}}(d')}, \tag{2}$$

where $\text{PPL}_{\text{p50}}(d)$ is the 50th percentile (median) of the perplexity distribution of dataset $d$, and $\rho_s$ is the global subset ratio. These ratios satisfy $\sum_{d \in \mathcal{D}} \beta_d = \rho_s$ and are used to form the initial dataset collection $\mathcal{D}'$ for training, prioritizing datasets with higher median perplexity values. During training, the $\text{PPL}_{\text{p50}}(d)$ values are efficiently updated using examples from the current batch, avoiding additional forward passes. The updated values adjust the dataset ratios $\beta_d$, increasing the proportion of datasets with higher $\text{PPL}_{\text{p50}}(d)$ and thereby shifting SST's focus towards more challenging tasks. (3) Define selection window parameters and initialize reference percentiles. For each dataset $D_d$ in the collection $\mathcal{D}$, the reference perplexity percentile, $P_d^{\text{ref}}$, is initialized to the 50th percentile. This $P_d^{\text{ref}}$ denotes the center of the dynamic selection window for $D_d$. The width of this selection window, in percentile points of that dataset's perplexity distribution, is given by $W_d = \beta_d \times 100$. Consequently, SST selects examples from $D_d$ whose perplexities fall within the percentile range $[P_d^{\text{ref}} - W_d/2, P_d^{\text{ref}} + W_d/2]$. During training, $P_d^{\text{ref}}$ is adaptively modified. It is constrained by the bounds $[W_d/2, 100 - W_d/2]$ (i.e., $[\frac{\beta_d \times 100}{2}, 100 - \frac{\beta_d \times 100}{2}]$) to ensure the selection window remains entirely within the 0–100th percentile range. This initialization strategy ensures that, at the beginning of the adaptive phase, the selection window focuses on the median difficulty examples within each dataset before any dynamic adjustments are made based on the training loss. (4) Set $k = t_w/t_{\max}$, where $t_w$ is the actual number of iterations completed during the warm-up phase. This value $k$ determines the rolling window size ($k \cdot t_{\max}$ iterations) for monitoring the loss curve during the adaptive scheduling phase. (5) Select a subset collection $\mathcal{D}'$ from $\mathcal{D}$ using $\beta_d$ and the reference perplexity percentiles $P_d^{\text{ref}}$. The main training loop runs for the remaining $t_{\max} - t_w$ iterations, ensuring that the total number of training iterations is $t_{\max}$, which also allows for a fair comparison with other methods and baselines. During this loop, SST tracks the training loss using a rolling window of size $k \cdot t_{\max}$ and updates the perplexity values of the examples in the current batch. Then, at every $k \cdot t_{\max}$ steps, SST carries out: (1) Recalculate $\beta_d$ (Equation 2). (2) Adjust the selection window based on the slope of the training loss curve. If the slope is negative (indicating decreasing loss), the current data configuration is likely not challenging enough. In this case, SST updates $P_d^{\text{ref}}$ by multiplying it by $(1 + \tau)$ (subject to the upper bound), shifting focus to harder examples. Conversely, if the slope is positive (indicating increasing loss), SST updates $P_d^{\text{ref}}$ by multiplying it by $(1 - \tau)$ (subject to the lower bound), shifting focus to easier examples. Otherwise, SST continues training without making any changes. (3) When $P_d^{\text{ref}}$ and $\beta_d$ change, SST samples $\mathcal{D}'$ using the updated values. When evaluating the slope, SST uses a small $\varepsilon = 10^{-3}$ to account for numerical instability. Further, SST caps $P_d^{\text{ref}}$ to ensure it remains within valid ranges. The value $\tau$ balances adaptability and stability. It is similar to a learning rate in optimization algorithms, controlling the rate of change in the selection window. In our experiments, we find $\tau = 0.1$ effective. We use a weighted sampler to dynamically prune examples by assigning their weight to 0. This approach enables the efficient exclusion and reintegration of examples into the training pool without incurring additional overhead from data loading. We present the Spaced Scheduled Training (SST) algorithm, providing a simplified overview in Algorithm 1 and the complete version in Algorithm 2. Key hyperparameters are listed in Table 1, and Table 5 summarizes the notations used throughout the description.

## 4.2 Overhead Analysis and Mitigation

As with any model-based data selection method, Sst introduces a computational overhead required to score the data. This overhead is often a concern in practice, especially at scale, as it can diminish a method's real-world usefulness. Training time alone is an insufficient metric for comparing overhead across methods because it depends on specific hardware and implementation details and may not fully capture scaling behavior related to the cost of scoring examples. Here, we propose a theoretical framework based on LLM scaling laws (Kaplan et al., 2020) to quantify the scaling of the overhead with the data and model sizes as a principled way to compare model-based data selection methods, and provide time measurements to provide a practical guideline for using Sst.

Following (Kaplan et al., 2020), given a dataset $D$, the computational cost $\mathcal{C}(\Pi_{rand})$ of training a model under a random sampling policy $\Pi_{rand}$ for $N$ epochs can be approximated as

$$\mathcal{C}(\Pi_{\text{rand}}) \approx N \cdot |D| \cdot (\mathcal{C}_{\text{forward}} + \mathcal{C}_{\text{backward}}) \approx N \cdot |D| \cdot 3\,\mathcal{C}_{\text{forward}}, \quad \text{where } \mathcal{C}_{\text{backward}} \approx 2\,\mathcal{C}_{\text{forward}}, \qquad (3)$$

with $\mathcal{C}_{\text{forward}}$ and $\mathcal{C}_{\text{backward}}$ being respectively the computational costs of a single forward and backward pass. Next, consider a model-based data selection method $\Pi_{\text{select}}$ requiring a single forward pass to evaluate the data for filtering (e.g., static pruning or Sst). Its training cost $\mathcal{C}(\Pi_{\text{select}})$ can be written as:

$$\mathcal{C}(\Pi_{\text{select}}) \approx \underbrace{\left(|D| \cdot \mathcal{C}'_{\text{forward}} + \mathcal{C}'_{\text{misc}}\right)}_{\text{selection cost}} + \underbrace{\left(\rho_s\, N \cdot |D| \cdot 3\,\mathcal{C}_{\text{forward}}\right)}_{\text{training cost on filtered data}} \approx \mathcal{C}'(\Pi_{\text{select}}) + \rho_s\,\mathcal{C}(\Pi_{\text{rand}}), \qquad (4)$$

where $\rho_s \in [0,1)$ is the subset ratio, $\mathcal{C}'_{\text{forward}}$ is the forward-pass cost of the reference model used for evaluation, $\mathcal{C}'_{\text{misc}}$ covers any additional overhead such as adjusting selection windows when using Sst, and $\mathcal{C}'(\Pi_{\text{select}})$ is the overhead cost when using a model-based method $\Pi_{\text{select}}$. A data selection method is considered computationally efficient if its overhead (the first term in Equation 4) is offset by the reduced training cost from using only $\rho_s|D|$ examples. Contrasting Equations 3 and 4, we conclude that $\Pi_{\text{select}}$ is efficient if $\mathcal{C}(\Pi_{\text{select}}) \leq \mathcal{C}(\Pi_{\text{rand}})$. Therefore, Sst is computationally efficient provided that

$$\mathcal{C}'(\Pi_{\text{select}}) + \rho_s\,\mathcal{C}(\Pi_{\text{rand}}) \leq \mathcal{C}(\Pi_{\text{rand}}) \quad \Longleftrightarrow \quad \rho_s \leq 1 - \frac{1}{3N} \quad \text{with } \mathcal{C}'_{\text{misc}} \approx 0. \qquad (5)$$

Sst uses the same model for reference and training, so $\mathcal{C}'_{\text{forward}} = \mathcal{C}_{\text{forward}}$. For simplicity, we assume $\mathcal{C}'_{\text{misc}} \approx 0$. This is because the miscellaneous costs (e.g., calculating the loss slope, updating dataset ratios $\beta_d$) are typically negligible compared to the cost of the initial perplexity computation for the entire dataset $(|D| \cdot \mathcal{C}'_{\text{forward}})$. Equation 5 also incorporates the warm-up phase (§ 4) (cf. Algorithm 2). Under these assumptions, we choose $\rho_s = 0.3$ for the experiments in § 5, comfortably satisfying Equation 5 for $N = 2$, ensuring Sst is computationally efficient. In Sst, the selection cost $\mathcal{C}'(\Pi_{\text{SST}})$ representing the overhead of the method, scales with the target model size, unlike methods relying on a constant-sized reference model $(\Pi_{\text{const}})$, such as InsTag (Lu et al., 2023), where the selection cost $\mathcal{C}'(\Pi_{\text{const}})$ is constant regardless of the target model size. To mitigate this overhead, inference-optimized frameworks can be used to accelerate perplexity computation.

To evaluate the potential speedup, we experimented with vLLM (Kwon et al., 2023) which provides a $1.8\times$ to $2.7\times$ speedup with Bfloat16, and further speedups with 8-bit or 4-bit quantization using model sizes from 0.5B to 32B and contrasted the speedup with InsTag using an 8B reference model using the ratio $\mathcal{C}'(\Pi_{\text{SST}})/\mathcal{C}'(\Pi_{\text{InsTag}})$. Since the downstream performance is likely to be affected when using lower precision, we also evaluate the performance using the evaluation setup described in §3 and contrasted each with the performance when using Bfloat16 precision without vLLM. We provide the details of the experimental setup in Appendix C.2. The results of this study in Figure 6 show that using an optimized inference framework like vLLM is able to offset the overhead introduced by Sst at larger model sizes. Specifically, we observe that when $\mathcal{C}'(\Pi_{\text{SST}})/\mathcal{C}'(\Pi_{\text{InsTag}}) \geq 1$ it becomes beneficial to use vLLM with 8-bit precision (Sst+ vLLM 8-bit) as it balances the overhead and performance trade-off. In contrast, using 4-bit precision (Sst+ vLLM 4-bit) introduces a non-negligible performance degradation. The main experimental results use a baseline environment (no vLLM) for fairness since some of the performance speedup in vLLM are highly dependent on the hardware configuration, but Figure 6 is an additional ablation.

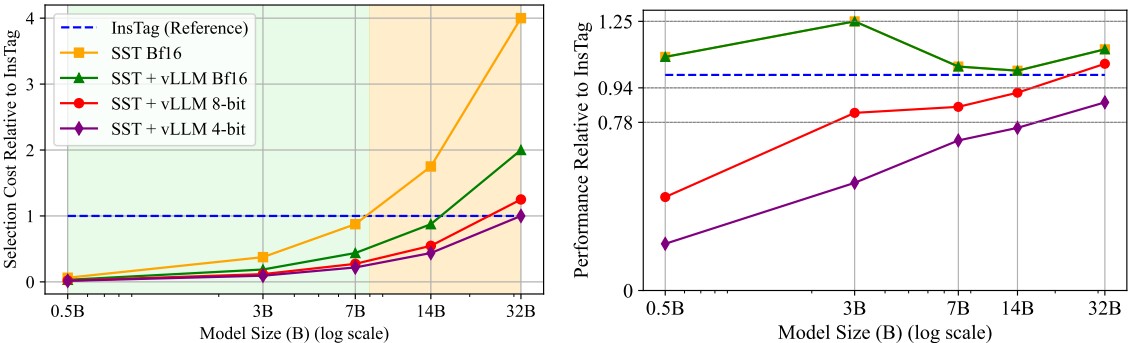

Figure 6: Overhead and performance analysis of Sst using vLLM with different floating-point precisions Sst compared to InsTag using a fixed 8B parameters reference model: Sst introduces significantly lower overhead for models smaller than 8B parameters even without vLLM (green region). For larger models (orange region), vLLM effectively offsets Sst's overhead, with 8-bit precision (in red) providing the best balance between overhead reduction and performance retention. (Left) Shows the scaling of the selection cost ratio $(\mathcal{C}'(\Pi_{\text{SST}})/\mathcal{C}'(\Pi_{\text{InsTag}}))$, defined above) with the model size. (Right) Shows the performance ratio of Sst with vLLM at different precision levels relative to InsTag.

Finally, we compare wall-clock times for training with and without Sst across three runs on three different model sizes (Appendix C). For instance, training the Qwen2.5-32B model on 100k examples takes about 36 hours, whereas using Sst to evaluate the 100k then train on only 30k examples takes about 17 hours. This 17-hour runtime includes approximately 34% overhead from Sst. Yet, it still cuts total training time by approximately 50% compared to the 36-hour baseline. These figures provide a rough estimate of the overhead introduced by Sst in practice. Moreover, the overhead is acceptable when $\rho_s$ satisfies the condition in Equation 5 and the performance $Q(\theta_{\Pi_{SST}})$ is either significantly better than $Q(\theta_{\Pi_{rand}})$ (as shown in §5) or remains comparable if compute resources are limited. In addition to the computational overhead, Sst requires storage for perplexity values over the dataset, i.e. $O(|D|)$. In our experimental setup (100k examples, *Bfloat16* precision), this memory footprint is under 0.2MB.

## 5 Experiments

This section outlines the experimental design used to evaluate Sst's effectiveness in the IFT setting. We describe the experimental setup (§5.1), present the results and discussion (§5.2), and conclude with limitations (§5.3).

### 5.1 Experimental Setup

We use the data, training settings, and evaluation procedures described in §3, consistent with the methodology of Lambert et al. (2024) (see Appendix A for details). Each method selects 30k examples from a 100k data pool, as our findings in §3 indicate that this subset is sufficient to match the performance of using the full dataset. We compare our method to Deita and InsTag, the current best-performing methods. We exclude methods like AlpaGasus, which require commercial models that prohibit rigorous experimentation due to cost constraints. Additionally, we do not compare against dynamic methods, as prior work (Liu et al., 2024; Lu et al., 2023; Kaddour et al., 2023) demonstrated that LLM-based static methods outperform existing dynamic methods in the IFT setting explored here. To ensure fairness, we restrict the pool to examples with fewer than 2048 tokens, as Deita and InsTag baselines rely on external tagging models with this token limit. We compare our method (Sst$_{30k}$) against: (1) Uniform$_{30k}$, which samples data uniformly across the dataset; (2) InsTag$_{30k}$, which uses the instruction-tagging method of Lu et al. (2023); (3) Deita$_{30k}$, which follows Liu et al. (2024) with complexity and quality scoring, and embedding-based diversity selection; (4) Rho-Loss$_{30k}$, which follows Mindermann et al. (2022); (5) Full$_{100k}$, which uses the entire pool without selection. For Sst$_{30k}$, we set $r_w = 3$, $\rho_s = 0.3$, $\tau = 0.1$, and $k_w = 0.1$ in all experiments, and compute perplexity only on target tokens to align with the IFT setting. For InsTag$_{30k}$ and Deita$_{30k}$, we use the

Table 2: Performance comparison of SST30$k$ with baseline methods across various model sizes and benchmarks. Results demonstrate that SST consistently outperforms baseline methods, including those using significantly more training data (FULL100$k$), in most scenarios. Notably, SST achieves superior performance on larger models (e.g., Qwen2.5-32B) and challenging benchmarks (e.g., MMLU-PRO), while maintaining competitive results on smaller models and diverse tasks.

| Method | Avg | MATH Lvl 5 | IFEval | MMLU-PRO | GPQA | MUSR | BBH |
|---|---|---|---|---|---|---|---|
| | | | | Qwen2.5-32B | | | |
| FULL$_{100k}$ | 39.58 | 34.67 | 73.9 | 48.03 | 16.33 | 16.52 | 48.06 |
| UNIFORM$_{30k}$ | 40.84 | 35.8 | 74.02 | 49.24 | _19.35_ | 19.14 | 47.51 |
| INSTAG$_{30k}$ | 38.21 | 35.42 | 70.01 | 46.51 | 17.11 | 15.43 | 44.74 |
| DEITA$_{30k}$ | 36.49 | 34.44 | 71.78 | 48.29 | 9.06 | 20.7 | 34.67 |
| RHO-LOSS$_{30k}$ | 38.56 | 30.31 | 72.10 | 48.03 | 16.00 | 16.93 | 48.00 |
| SST$_{30k}$ (ours) | _42.75_ | _36.1_ | _75.03_ | _53.25_ | 18.46 | _24.79_ | _48.85_ |
| | | | | G2-27b | | | |
| FULL$_{100k}$ | 32.34 | 21.68 | _72.26_ | 32.18 | 11.41 | 16.84 | _39.68_ |
| UNIFORM$_{30k}$ | 30.63 | 19.86 | 70.16 | 28.8 | 12.08 | 15.24 | 37.63 |
| INSTAG$_{30k}$ | 31.24 | 22.66 | 68.01 | 31.38 | 10.63 | 17.77 | 36.98 |
| DEITA$_{30k}$ | 32.07 | _23.11_ | 71.93 | _35.97_ | 8.5 | 16.78 | 36.11 |
| RHO-LOSS$_{30k}$ | 27.48 | 20.45 | 67.93 | 26.89 | 4.50 | 12.47 | 32.66 |
| SST$_{30k}$ (ours) | _32.89_ | 20.92 | 70.28 | 34.28 | _12.64_ | _19.66_ | 39.55 |
| | | | | G2-9b | | | |
| FULL$_{100k}$ | 28.43 | _15.26_ | 66.74 | 29.52 | 8.95 | 18.58 | 31.56 |
| UNIFORM$_{30k}$ | 28.85 | 11.93 | 65.36 | _30.8_ | 9.96 | 21.56 | 33.47 |
| INSTAG$_{30k}$ | 28.58 | 12.39 | 66.97 | 29.14 | _10.85_ | 17.59 | _34.54_ |
| DEITA$_{30k}$ | 28.08 | 14.27 | _67.58_ | 28.3 | 8.95 | 17.26 | 32.11 |
| RHO-LOSS$_{30k}$ | 26.79 | 13.00 | 64.11 | 26.56 | 10.41 | 14.42 | 32.22 |
| SST$_{30k}$ (ours) | _29.17_ | 11.48 | 66.39 | 30.5 | 10.29 | _23.33_ | 33.02 |
| | | | | Llama3.1-8B | | | |
| FULL$_{100k}$ | _23.59_ | _5.44_ | _62.41_ | 23.39 | 4.81 | _15.97_ | _29.53_ |
| UNIFORM$_{30k}$ | 21.11 | 5.29 | 61.85 | 21.8 | _5.7_ | 15.32 | 16.71 |
| INSTAG$_{30k}$ | 21.61 | 5.06 | 55.54 | _24.75_ | 4.81 | 14.31 | 25.2 |
| DEITA$_{30k}$ | 18.02 | 4.91 | 59.27 | 20.77 | 0.0 | 9.0 | 14.15 |
| RHO-LOSS$_{30k}$ | 19.77 | 4.76 | 48.89 | 22.93 | 4.60 | 14.17 | 23.27 |
| SST$_{30k}$ (ours) | 22.45 | 4.53 | 59.28 | 23.84 | 4.25 | 14.63 | 28.19 |
| | | | | Llama3.2-3B | | | |
| FULL$_{100k}$ | 14.88 | 1.44 | 46.93 | 15.58 | 2.01 | 6.5 | _16.8_ |
| UNIFORM$_{30k}$ | 15.52 | 1.89 | _49.95_ | 14.33 | _3.91_ | 9.03 | 13.99 |
| INSTAG$_{30k}$ | 13.08 | 1.81 | 43.43 | 13.24 | 1.23 | 9.16 | 9.62 |
| DEITA$_{30k}$ | 11.73 | 1.66 | 46.54 | 12.08 | 0.0 | 2.73 | 7.36 |
| RHO-LOSS$_{30k}$ | 10.73 | _2.00_ | 39.39 | 9.13 | 2.68 | 3.17 | 8.03 |
| SST$_{30k}$ (ours) | _16.33_ | 1.59 | 49.13 | _16.26_ | 3.58 | _11.24_ | 16.2 |
| | | | | Llama3.2-1B | | | |
| FULL$_{100k}$ | _7.45_ | 0.45 | _34.96_ | _1.66_ | _0.0_ | _3.62_ | 4.01 |
| UNIFORM$_{30k}$ | 6.59 | 0.08 | 33.56 | 1.0 | _0.0_ | 2.38 | 2.55 |
| INSTAG$_{30k}$ | 6.8 | 0.23 | 31.61 | 1.52 | _0.0_ | 3.17 | 4.3 |
| DEITA$_{30k}$ | 6.46 | 0.38 | 29.51 | 1.22 | _0.0_ | 2.93 | 4.76 |
| RHO-LOSS$_{30k}$ | 5.74 | _0.60_ | 27.64 | 1.29 | _0.0_ | 2.14 | 2.76 |
| SST$_{30k}$ (ours) | 6.56 | 0.15 | 29.18 | 1.49 | _0.0_ | 3.52 | _5.01_ |
| | | | | Qwen2.5-0.5B | | | |
| FULL$_{100k}$ | 7.32 | 0.91 | _31.07_ | 5.88 | 0.22 | 1.27 | _4.56_ |
| UNIFORM$_{30k}$ | 7.05 | 0.91 | 29.09 | 6.65 | 0.11 | _1.43_ | 4.14 |
| INSTAG$_{30k}$ | 6.94 | 1.21 | 28.88 | 6.37 | 0.45 | 1.11 | 3.66 |
| DEITA$_{30k}$ | 6.81 | 1.21 | 28.61 | _6.83_ | 0.0 | _1.43_ | 2.76 |
| RHO-LOSS$_{30k}$ | 5.92 | _1.81_ | 24.00 | 5.00 | 0.00 | 1.02 | 3.67 |
| SST$_{30k}$ (ours) | _7.52_ | 0.98 | 29.18 | 6.56 | _3.02_ | 1.27 | 4.12 |

original scoring models from the respective works. For RHO-LOSS$_{30k}$, we adapt the original method[3] to the IFT setting and set the number of training iterations to match the number of training steps in SST$_{30k}$. We report all results in Table 2.

---

[3]https://github.com/OATML/RHO-Loss

## 5.2 Results and Discussion

The results in Table 2 demonstrate the clear advantages of our proposed method over baseline approaches. Using only 30% of the training data, $\text{SST}_{30k}$ achieves significant performance improvements across different models. It surpasses other baselines, including the one trained on the full 100k dataset in five out of seven tested models and closely matches the performance on the remaining model (Llama3.2-1B) compared to methods utilizing the same amount of data. These results highlight SST's effectiveness in optimizing computational resources. The observation that SST, utilizing only a data subset (e.g., $\text{SST}_{30k}$), can outperform baselines trained on the full dataset ($\text{FULL}_{100k}$) in several cases (Table 2) extends beyond merely filtering erroneous examples and relates to data efficiency. Even curated instruction mixtures like the TÜLU 3 mixture (Lambert et al., 2024) may contain redundancies or examples offering diminishing returns, making subsets potentially more compute-effective. Indeed, Lambert et al. (2024) demonstrated that subsets (e.g., 25-75%) of this mix often performed comparably to the full set within a fixed budget, echoing findings where smaller, highly curated sets proved highly effective (Zhou et al., 2023a). Furthermore, SST's adaptive nature allows it to tailor data selection to the target model's specific pre-training composition and evolving state, unlike static mixes which might oversample topics already mastered during pre-training. SST may also implicitly filter examples that are overly complex relative to a model's capabilities (e.g., complex reasoning for smaller models) or out-of-distribution relative to pre-training or evaluation data (e.g., low-resource languages not present in benchmarks, cf. Table 4.), thereby focusing training on more learnable and relevant content. Our dynamic data scheduling approach ensures a balanced exposure to examples of varying complexity tailored to the target model, enabling robust generalization across diverse tasks and models. In contrast, methods like $\text{DEITA}_{30k}$, which involve an elaborate data selection process or $\text{INSTAG}_{30k}$, which tries to replicate state-of-the-art models filtering behavior, fail to adapt to the model's evolving needs during training. These methods produce inconsistent results across the different model sizes and families. The results indicate that the adaptive baseline ($\text{RHO-LOSS}_{30k}$) exhibits inconsistent performance across different models and benchmarks compared to other methods. This general inconsistency aligns with previous findings regarding RHO-Loss (Kaddour et al., 2023). Interestingly, on the Math Lvl 5 benchmark specifically, it displays a noticeable trend of achieving higher scores on smaller models (Llama3.2-3B, Llama3.2-1B, and Qwen2.5-0.5B). Despite this specific strength, $\text{RHO-LOSS}_{30k}$ is significantly outperformed by the proposed by $\text{SST}_{30k}$ on average across all models and benchmarks. It is worth noting that while the RHO-Loss method was adapted for the IFT setting in this study, its observed inconsistency echoes the performance limitations reported by Kaddour et al. (2023) in the original context.

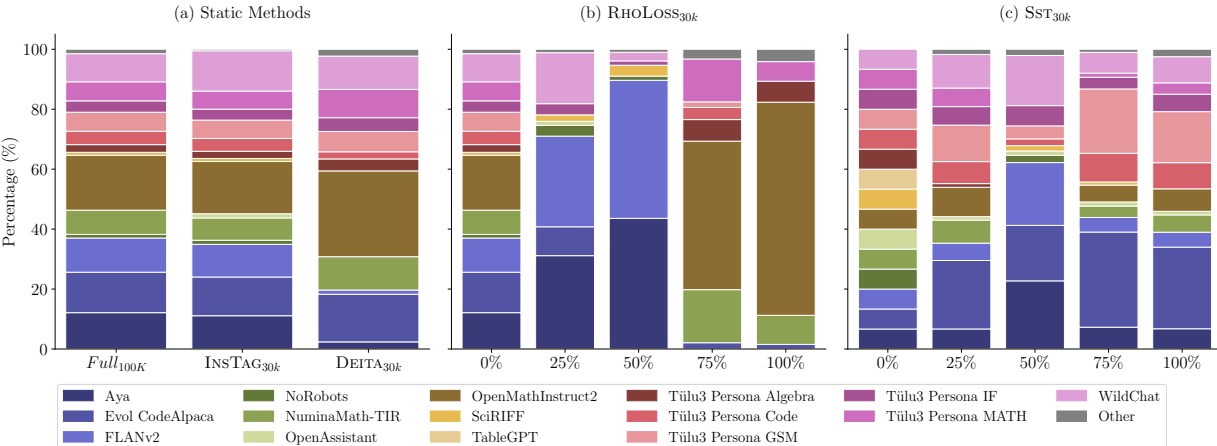

Figure 7: Comparison of data mixture composition for static and dynamic selection methods using Qwen2.5-32B. The figure shows the percentage distribution of datasets within the training data mixture for different selection baselines. Other represents the datasets that account for less than 1% of the total data. (a) Illustrates the fixed composition for static methods. (b) and (c) Depict the evolving data mixture composition for the dynamic methods $\text{RHO-LOSS}_{30k}$ and $\text{SST}_{30k}$, respectively, at various training progress points (0%, 25%, 50%, 75%, and 100%).

Figure 7 reveals distinct dynamic data selection patterns. RHO-LOSS (7b) increasingly sacrifices diversity, becoming heavily dominated by OpenMathInstruct2 and Tulu3 Persona MATH towards the end; this skewed focus potentially links to its overall inconsistency despite its observed tendency for better Math Lvl 5 performance specifically on smaller models. In contrast, SST (7c) appears to maintain a broader representation of datasets throughout training. This sustained diversity likely contributes to SST's more robust and consistently superior performance (Figure 1, Table 2)), contrasting with RHO-LOSS's inconsistency stemming from its lack of diversity, a crucial factor in IFT as shown in DEITA and INSTAG. Conversely, SST's mid-training shift towards Aya and FLAN V2 coincides with the temporary performance dip seen in Figure 1. Crucially, SST's subsequent recovery from this dip highlights its strong adaptability, demonstrating its ability to navigate transiently challenging data mixes and effectively refine selection away from the initial static distribution towards a more tailored mix, which likely underlies its sustained upward performance trend towards the end of training, reinforcing the SST's argument that adaptive selection based on the model's state leads to more effective data compositions than static approaches.

**Sst shows significant improvement using large models:** In §3, we showed that models with more than 20B parameters are less sensitive to static data selection methods. In contrast, our results show the efficacy of SST in leveraging smaller, carefully scheduled data subsets to maximize performance even for larger models. Using Qwen2.5-32B, $SST_{30k}$ delivers an average performance of 42.75%, surpassing all baselines including the one using more than threefold the amount of data ($FULL_{100k}$). Notably, the method improves results on challenging tasks such as MMLU-PRO, where it achieves a score of 53.25%, outperforming the $FULL_{100k}$ baseline by 5.22%. The G2-27B model further illustrates the benefits of our method, where $SST_{30k}$ achieves an average performance of 32.89%, outperforming $DEITA_{30k}$ and $INSTAG_{30k}$ while maintaining competitive results against $FULL_{100k}$. This trend persists across multiple tasks, with $SST_{30k}$ showing resilience and adaptability in both general benchmarks like GPQA and domain-specific tasks such as MATH LvL 5. Using Llama3.1-8B, $SST30k$ outperforms all baselines with comparable data sizes but falls behind $FULL_{100k}$. This gap may be due to the model's weaker instruction-following abilities, as seen in its lower IFEval benchmark scores. The earlier Llama3.1 likely lacked sufficient exposure to instruction-like examples during late-stage pre-training—a strategy used in newer models like Llama3.2 and Qwen2.5. To test this hypothesis, we increased the sample size from 30k to 50k examples. As shown in Table 3, $SST_{50k}$ outperforms all methods, including $FULL_{100k}$. This suggests that Llama3.1-8B benefits from additional data for optimal performance. While increasing the subset size improves results, an ideal approach would determine the optimal subset size rather than treating it as a fixed hyperparameter. We leave this exploration for future work.

Figure 1 shows the performance of $SST_{30k}$ compared to other methods on Qwen2.5-32B throughout the training process across four random seeds (42, 123, 456, 789), using a multi-seed evaluation strategy similar to Lambert et al. (2024). The figure highlights that the performance advantage of $SST_{30k}$ is not merely a final outcome but is maintained consistently throughout the training process compared to all baseline methods. Notably, $SST_{30k}$ exhibits lower variance compared to the other methods, indicating more stable and reliable training dynamics. Furthermore, the distinct upward trend observed for $SST_{30k}$ in the later training stages suggests that its lead might increase further with extended training beyond the two epochs evaluated. Further examination of the data mixture compositions, as depicted in Figure 7, reveals distinct behaviors among the selection strategies. Static methods like DEITA exhibit a fixed, less diverse subset, dominated by a few datasets compared to the Full

In analyzing the Llama-3.1-8B results (Table 2, Table 3), we note that while SST significantly improves the average leaderboard score compared to the base model, its performance on specific knowledge-intensive benchmarks (GPQA, MMLU-PRO) is lower. This aligns with the known challenge of knowledge degradation or "catastrophic forgetting" sometimes observed after SFT (Zheng et al., 2025). SFT may inadvertently overwrite pre-trained knowledge, particularly if the SFT dataset lacks sufficient explicit reinforcement of that knowledge. Supporting this, degradation on knowledge benchmarks (e.g., TruthfulQA, MMLU, GPQA) was observed when using larger SFT data fractions in Lambert et al. (2024). While SST exhibits this effect, similar degradation occurs with other sampling methods, and it may be pronounced here as our initial data pool did not specifically include tasks designed to reinforce this type of pre-trained knowledge.

Table 3: Performance comparison of $\textsc{Sst}_{50k}$ with the same baselines as in Table 2 on Llama3.1-8B using 50k examples instead of 30k. $\textsc{Sst}_{50k}$ outperforms all baselines, including $\textsc{Full}_{100k}$. This suggests that, for some models, the subset size is a critical hyperparameter for data selection.

| Method | Avg | MATH Lvl 5 | IFEval | MMLU-PRO | GPQA | MUSR | BBH |
|---|---|---|---|---|---|---|---|
| $\textsc{Full}_{100k}$ | 23.59 | 5.44 | 62.41 | 23.39 | 4.81 | 15.97 | 29.53 |
| $\textsc{Uniform}_{50k}$ | 21.35 | 5.37 | 61.23 | 22.22 | 5.70 | 15.00 | 18.55 |
| $\textsc{InsTag}_{50k}$ | 22.08 | 5.00 | 57.90 | 24.12 | 4.41 | 14.96 | 26.10 |
| $\textsc{Deita}_{50k}$ | 18.07 | 4.76 | 60.12 | 21.78 | 0.00 | 8.00 | 13.77 |
| $\textsc{Sst}_{50k}$ (ours) | 23.85 | 5.37 | 62.71 | 24.82 | 5.64 | 14.97 | 29.60 |

**Using intermediate tagger models can be detrimental:** Both $\textsc{Deita}$ and $\textsc{InsTag}$ rely on tagger models to select data to replicate the behavior of much larger models (e.g., ChatGPT). While this reduces evaluation costs, our results reveal significant performance drawbacks. For instance, $\textsc{Deita}_{30k}$ and $\textsc{InsTag}_{30k}$ lag behind the simple $\textsc{Uniform}_{30k}$ baseline on Qwen2.5-32B, because their tagger models struggle with highly complex examples (e.g., NuminaMath datasets) or sequences approaching the 2048-token limit—the limit of all tagging models proposed by $\textsc{Deita}$ and $\textsc{InsTag}$. This limitation is more noticeable for $\textsc{Deita}_{30k}$ with Qwen2.5-32B, likely because it relies on two taggers and an embedding model as opposed to a single tagger used by $\textsc{InsTag}$, which can further exacerbate the issue. On G2-27B, both methods show a comparable performance to other baselines, however, our results in §3 show that this model is particularly less sensitive to data selection methods, suggesting, in this setup, that the performance decreases with the complexity of the data selection approach, matching the findings of Marion et al. (2023) in the context of pre-training. This effect is also noticeable on smaller models such as Llama3.2-3B, where $\textsc{Uniform}_{30k}$ outperforms $\textsc{InsTag}_{30k}$ and $\textsc{Deita}_{30k}$ by 2.44 and 3.79 points, respectively. Our analysis suggests that the data selected by $\textsc{Deita}_{30k}$ and $\textsc{InsTag}_{30k}$ contains a significant number of complex examples for such models when contrasting their selection with perplexity-based categorization.

Table 4: FLAN V2 example in Malayalam selected by $\textsc{Deita}$. While the example is of a good quality, it may be detrimental for models not exposed to Malayalam data during pre-training, where such an example falls within the high-perplexity range. $\textsc{Sst}$ dynamically adjusts the selection window away from these examples.

---

**Inst**: You are given a statement written in Malayalam. (...) Output the word from the correct option.

`<MASK>` സർക്കാർ `1950` ആരംഭിച്ചതാണ് ഈ കേന്ദ്രം. പെരിന്തൽമണ്ണ കേന്ദ്രീകരിച്ചുള്ള മൃഗസംരക്ഷണ വികസനപദ്ധതിക്കു കീഴിലായിരുന്നു `(...)`

**A:** മദിരാശി **B:** പാലക്കാട് **C:** തിരുവിഴാംകുന്ന് **D:** പെരിന്തൽമണ്ണ

---

Further, tagger models face other limitations, such as their context window limitation, or when the tagger training data distribution differs from the distribution of the data that needs to be selected. The latter is particularly problematic in our experiment when evaluating multi-lingual data, as shown in Table 4. In both cases, the tagger weaknesses introduce additional noise in the data selection process, leading to suboptimal performance. Solving the issues would require training newer tagging models, inducing a significant cost, which questions the practicality of such methods.

## 5.3 Limitations and Future Work

$\textsc{Sst}$ remains effective as long as each dataset's perplexity distribution has a nontrivial overlap with the selection window. However, if the data is extremely unbalanced, simpler approaches (e.g., per-dataset random sampling) may perform better. Observations in § 3 show that a skewed perplexity distribution can lead to many noisy examples in the selection, and excluding them improved performance. Therefore, one could develop an $\textsc{Sst}$ variant that ignores these outliers or uses a different usefulness metric along with our modulation method. We did not explore such variations, leaving them for future research. Although we focused on IFT, $\textsc{Sst}$ could benefit pre-training as well. In this context, two challenges arise: (1) it is unclear if our warm-up triggers are suitable for pre-training, and (2) the overhead may be prohibitive with larger

datasets. A potential solution is to pre-compute perplexities offline using an external model—a standard approach in pre-training data cleaning (Penedo et al., 2023)—and then apply adaptive selection after a warm-up phase. Using 4-bit precision for perplexity evaluation could further reduce scoring costs, but its impact on selection quality in pre-training remains unclear. Nonetheless, the results of Marion et al. (2023) in pre-training, alongside our own findings in IFT, suggest that SST may offer significant gains in pre-training as well. We leave an in-depth investigation of these trade-offs to future work. In § 4.2, we assumed $\mathcal{C}_{\mathrm{misc}} \approx 0$ since we found this term negligible in our single-node experiments by syncing the gathering of perplexity values with gradient updates, to avoid additional inter-GPU synchronization. However, communication and synchronization overhead can grow significantly especially if high-speed interconnects like InfiniBand are not available or if the implementation does not take into account these factors.

## 6 Conclusion

Our proposed Spaced Scheduled Training (SST) framework provides an adaptive, efficient, and model-specific approach to data selection, eliminating reliance on external oracle models. We demonstrate that continuously adjusting the dataset composition based on real-time perplexity signals improves performance. Through extensive evaluations on seven LLMs (0.5B–32B) and a theoretical overhead analysis grounded in scaling laws, SST consistently achieves performance gains across architectures and scales efficiently to large training regimes. These results offer robust empirical evidence and practical insights for enhancing data quality in LLM instruction fine-tuning.

## Broader Impact Statement

Data selection methods like Spaced Scheduled Training (SST) enhance Large Language Model (LLM) performance and efficiency. However, any data sampling method, including SST, carries the risk of introducing or amplifying biases and ethical concerns (Mehrabi et al., 2022; Wang et al., 2025; Guo et al., 2024). LLMs are trained on extensive text datasets that often reflect societal biases (e.g., gender, race). Sampling methods, even those based on model confidence, may inadvertently favor or reinforce patterns from dominant groups in the data, potentially amplifying these biases (Wang et al., 2025). This can result in unfair outcomes, such as marginalizing perspectives in content generation or producing discriminatory results in decision-making systems. Defining and measuring fairness and bias remains challenging (Wang et al., 2025), particularly since bias is often deeply embedded in training data (Guo et al., 2024). Benchmarks like Helm Safety (Liang et al., 2023) can identify some issues but have limitations. Model performance is sensitive to benchmark design choices (e.g., prompt format, example order) (Grattafiori et al., 2024b), and evaluation data may be contaminated by pre-training data, potentially inflating scores and complicating accurate bias assessment (Grattafiori et al., 2024b). Effectively analyzing data contamination remains an open research challenge (Grattafiori et al., 2024b). In practice, LLMs are components of larger systems. Practitioners should implement system-level safety measures alongside existing benchmarks, often using external models to supplement model-level safeguards like Llama Guard (Grattafiori et al., 2024b). We acknowledge the importance of conducting a dedicated, large-scale analysis of the fairness implications and potential biases introduced or mitigated by perplexity-based sampling. We leave this exploration for future work.

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

## A    Experimental Setup Details

**Training Data Mixture:**  We use a stratified subsample of 100k examples from the recent Tulu 3 SFT Mix(Lambert et al., 2024) containing 15 datasets across diverse tasks and domains. These include FLAN v2 (Longpre et al., 2023), No Robots (Rajani et al., 2023), OpenAssistant (Köpf et al., 2023), Tulu 3 Persona MATH, Tulu 3 Persona GSM, Tulu 3 Persona Python, Tulu 3 Persona Algebra, Tulu 3 Persona IF (Lambert et al., 2024), NuminaMath-TIR (LI et al., 2024), Aya (Singh et al., 2024), WildChat GPT-4 (Zhao et al., 2024), TableGPT (Li et al., 2023), SciRIFF (Köpf et al., 2023), Evol CodeAlpaca (Luo et al., 2023). Using 10% of the full mixture allows us to perform rigorous experimentation across multiple models while requiring reasonable amount of compute, allowing for reproducibility by future research. Further, (Lambert et al., 2024) showed minimal average performance drop even with a 5% subset of the Tulu 3 mixture.

**Training and Evaluation Setup:**  We use the same training setup and code base proposed by (Lambert et al., 2024). We perform full parameter training for two epochs with an effective batch size of 128, a learning rate (LR) of $5e-06$ using a linear LR scheduler with a 3% warm-up ratio. We set the maximum sequence length to 2048. All models were trained on 8 NVIDIA H100 GPUs using FlashAttention 2 (Dao, 2024) and DeepSpeed Zero-Stage 3 (Rajbhandari et al., 2019). For models larger than 20B we use the 32bit paged Adam optimizer. We provide the full training setup in our public repository[4]. We evaluate our models using the Open LLM Leaderboard 2 (Fourrier et al., 2024) tasks. It addresses performance saturation issues from the earlier version (Beeching et al., 2023) by introducing harder and less contaminated benchmarks. This update enables more meaningful evaluation result, particularly for recent LLMs which is crucial for our experiment. It includes IFEval (Zhou et al., 2023b), BBH (Suzgun et al., 2022), MATH LvL 5 (Hendrycks et al., 2021), GPQA (Rein et al., 2023), MuSR (Sprague et al., 2024), and MMLU-PRO (Wang et al., 2024). We ensure reproducibility by using the same lm-evaluation-harness (Gao et al., 2021) version and report the average normalized scores across all benchmarks as in (Fourrier et al., 2024).

**Methods and Baselines:**  Similar to Marion et al. (2023), we train models on 10%, 30%, and 50% of the data from the bottom, middle, and top segments of the per-example perplexity distribution. For instance, to create a 10% middle subsample, we select examples between the 45th and 55th percentiles. For each subset size, we create two datasets one using the overall perplexity distribution of the mixture and the other using the per-dataset perplexity distribution. To select data, we use either the pre-trained version of the target model as a reference, or checkpoints of the same model trained on same subset at different train iteration. Specifically, we select data using checkpoints at 0.25, 0.5, 1, and 2 epochs. When using the pre-trained model as reference, we use a simple chat template proposed by (Lambert et al., 2024) to encode the chat data, where we prepend each turn content with its role (e.g., "User: ") and separate the turn with a new line. This allows us to avoid adding special tokens to the tokenizer which will require resizing the model's embedding layer. We compare the methods to various baselines where we train the same pre-trained model on: (1) the full data mixture (100%), (2) a random subset of the same size (Random), (3) and a random subset drawn uniformly for each dataset (Uniform). For the random and uniform baselines, show the average performance across the two random seeds (123, and 42) and report the standard error.

**Models:**  To better study the impact of the perplexity-based data selection across different model characteristics, and to ensure that our findings are generalizable, we use a diverse set of models. Specifically, we use different model families including Llama3.1 (L3.1), Llama3.2 (L3.2), Qwen2.5 (Q2.5), and Gemma2 (G2). This allows us to study how pre-training factors, such as the pre-training data composition and training setting affect the data selection performance. Further, we compare our models a cross different models sizes ranging from 0.5B to 32B parameters, to study the influence of model size. In total, we trained seven models each with 31 different training configurations, resulting in 248 training runs for this experiment.

---

[4]Available after double-blind review.

# B   Method Details

---

**Algorithm 2** Spaced Scheduled Training (SST)

---

**Require:** $\theta$: initial model parameters; $\mathcal{D} = \{D_1, \ldots, D_d\}$: collection of $d$ datasets; $t_{\max}$: maximum training iterations; $k_w \in (0,1]$: warm-up slope window ratio; $r_w$: maximum warm-up slope evaluation retry count. $\rho_s \in (0,1]$: global subset ratio; $\tau \in (0,1]$: window-shift ratio; $\varepsilon > 0$: slope threshold for loss slope evaluation;
**Ensure:** Updated model parameters $\theta$.

1: **Algorithm:** SST($\theta, \mathcal{D}, t_{\max}, \rho_s, \tau, \varepsilon, r_w, k_w$)

2: **Warm-up phase**
3: $\theta, t_w \leftarrow$ WARMUP($\theta, \mathcal{D}, t_{\max}, r_w, k_w$)

4: **Adaptive Scheduling Phase**
5: $k \leftarrow t_w / t_{\max}$
6: $P \leftarrow$ COMPUTEPPL($\theta_{t_w}, \mathcal{D}$)  ▷ Compute per-example perplexities PPL($e$) for each example $e \in \mathcal{D}$
7: Compute $\beta_d$ for each dataset $D_d$ using Equation 2
8: $P_d^{\text{ref}} \leftarrow 50$ for each dataset $D_d$  ▷ Initialize window reference to the 50th percentile
9: $\mathcal{D}' \leftarrow$ SELECTPERDATASETSUBSET($\mathcal{D}, \rho_s, \{\beta_d\}, \{P_d^{\text{ref}}\}$)
10: Initialize loss buffer $L_{\text{buf}} \leftarrow []$ (capacity $\lfloor k \cdot t_{\max} \rfloor$)
11:  ▷ Main training loop
12: **for** $t = t_w + 1$ **to** $t_{\max}$ **do**
13:     $\theta, L_B, P \leftarrow$ TRAINSTEP$\left(\theta, \text{SAMPLEBATCH}(\mathcal{D}')\right)$  ▷ Train model on a batch, and update perplexities
14:     Append $L_B$ to $L_{\text{buf}}$
15:     **if** $|L_{\text{buf}}| = \lfloor k \cdot t_{\max} \rfloor$ **then**
16:         $slope \leftarrow$ COMPUTESLOPE($L_{\text{buf}}$)  ▷ e.g., via simple linear regression
17:         Clear $L_{\text{buf}}$
18:  ▷ 1) Recompute dataset mixing ratios
19:         Update $\beta_d$ for each dataset $D_d$ using Equation 2
20:  ▷ 2) Decide if window shifts to harder or easier examples
21:         **if** $slope < -\varepsilon$ **then**
22:             $P_d^{\text{ref}} \leftarrow \min\left(P_d^{\text{ref}} \times (1 + \tau),\ 100 - \frac{\beta_d \times 100}{2}\right)$  ▷ Loss is decreasing, shift to more complex examples
23:         **else if** $slope > \varepsilon$ **then**
24:             $P_d^{\text{ref}} \leftarrow \max\left(P_d^{\text{ref}} \times (1 - \tau),\ \frac{\beta_d \times 100}{2}\right)$  ▷ Loss is increasing, shift to easier examples
25:         **else**
26:             **no change to** $P_d^{\text{ref}}$  ▷ Loss is stable, keep the current window
27:         **end if**
28:  ▷ 3) Select updated subset based on new mixing ratios and reference perplexity percentiles
29:         $\mathcal{D}' \leftarrow$ SELECTPERDATASETSUBSET($\mathcal{D}, \rho_s, \beta_d, P_d^{\text{ref}}$)
30:     **end if**
31: **end for**
32: **return** $\theta$  ▷ Final trained model parameters

33: **function** WARMUP($\theta, \mathcal{D}, t_{\max}, r_w, k_w$)
34:     $t_w \leftarrow 0$
35:     $r \leftarrow 0$
36:     Initialize a loss buffer $L_{\text{buf}} \leftarrow []$ (capacity $\lfloor k_w \cdot t_{\max} \rfloor$)
37:     **while** $retry < r_w$ **do**
38:         $\theta, L_B \leftarrow$ TRAINSTEP$\left(\theta, \text{UNIFORMSAMPLE}(\mathcal{D})\right)$
39:         Append $L_B$ to $L_{\text{buf}}$
40:         **if** $|L_{\text{buf}}| = \lfloor k_w \cdot t_{\max} \rfloor$ **then**
41:             $slope \leftarrow$ COMPUTESLOPE($L_{\text{buf}}$)
42:             Clear $L_{\text{buf}}$
43:             **if** $-\varepsilon \leq slope \leq \varepsilon$ **then**  ▷ Loss stabilized
44:                 **break**  ▷ End warm-up early
45:             **end if**
46:         **end if**
47:         $t_w \leftarrow t_w + 1$
48:         $r \leftarrow r + 1$
49:     **end while**
50:     **return** $\theta, t_w$
51: **end function**

52: **function** SELECTPERDATASETSUBSET($\mathcal{D}, \rho_s, \beta_d, P_d^{\text{ref}}$)
53:     **for** $d = 1$ **to** $m$ **do**
54:         $P_{low} \leftarrow P_d^{\text{ref}} - \frac{\beta_d \times 100}{2}, \quad P_{high} \leftarrow P_d^{\text{ref}} + \frac{\beta_d \times 100}{2}$
55:         $\mathcal{D}'_d \leftarrow \{\, e \in D_d : P_{low} \leq \text{PPL}(e) \leq P_{high}\}$  ▷ Keep examples whose perplexities fall within [low, high]-th percentile
56:     **end for**
57:     $\mathcal{D}' \leftarrow \bigcup_{d=1}^{m} \mathcal{D}'_d$
58:     **return** $\mathcal{D}'$
59: **end function**

To clarify the methodology presented, we now define the key notations used throughout our description of the Spaced Scheduled Training (SST) algorithm in Table 5 which provides a comprehensive summary of these symbols and their corresponding definitions for easy reference.

Table 5: Summary of key notations and symbols used in the Spaced Scheduled Training (SST) algorithm.

| Notation | Description |
|---|---|
| | General Training Parameters |
| $t_{\max}$ | Maximum number of training iterations. |
| $\rho_s$ | Global subset ratio (fraction of data to select). |
| $\mathcal{D}$ | Collection of datasets $\{D_1, \ldots, D_d\}$. |
| $\mathcal{D}'$ | Selected subset collection from $\mathcal{D}$ used for training. |
| | Perplexity Calculation |
| $PPL(e)$ | Per-example perplexity for example $e$. |
| $PPL_{p50}(d)$ | The 50th percentile (median) perplexity for dataset $D_d$. |
| | Warm-up Phase Parameters |
| $t_w$ | Number of iterations spent in the warm-up phase. |
| $k_w$ | Warm-up slope evaluation window size ratio (fraction of $t_{\max}$). |
| $r_w$ | Maximum warm-up retry count for slope evaluation. |
| | Adaptive Phase - Selection Window Control |
| $\beta_d$ | Dataset mix ratio for dataset $D_d$. |
| $P_d^{ref}$ | Reference perplexity percentile for dataset $D_d$. |
| $P_{low}, P_{high}$ | Lower/upper percentile bounds for selection window. |
| $\tau$ | Selection window adjustment factor (shift ratio). |
| $k$ | Rolling window size ratio for loss monitoring (fraction of $t_{\max}$). |
| $\epsilon$ | Slope threshold for loss stability evaluation. |

## C   Overhead

### C.1   Wall-Clock Time Comparison

In this section, we present empirical measurements of the overhead introduced by Sst compared to random sampling. We use the 100k dataset described in Section 3. For random sampling, we train on all 100k examples, whereas for Sst, we select a 30k subset out of the same 100k examples and train on that subset following the same training setup as in Section 3. To quantify the overhead, we define the wall-clock time for each method as $\mathcal{T}(\cdot)$ and estimate the relative overhead ratio

$$\mathcal{T}_{\text{overhead}} = \frac{\mathcal{T}(\varPi_{\text{SST}_{30k}}) - \mathcal{T}(\varPi_{\text{rand}_{30k}})}{\mathcal{T}(\varPi_{\text{SST}_{30k}})}, \tag{6}$$

where $\mathcal{T}(\varPi_{\text{SST}_{30k}})$ is the measured time to train using Sst on the 30k subset, and $\mathcal{T}(\varPi_{\text{rand}_{30k}}) = 0.3\mathcal{T}(\varPi_{\text{rand}_{100k}})$ is the time taken by random sampling on 30k examples. We approximate $\mathcal{T}(\varPi_{\text{rand}_{30k}})$ by scaling down the measured 100k run time, assuming per-example costs remain roughly constant.

Table 6 compares the training times on 100k examples (random sampling) versus Sst on a 30k subset. Although the overhead of evaluating and filtering the data is significant, the training time reduction is substantial: for instance, training Qwen2.5-32B is reduced from 36.7 hours to 16.8 hours on our hardware. These numbers are highly dependent on hardware configuration and implementation details, but they provide a rough estimate of Sst's overall savings. On a multi-node setup, this overhead may increase due to additional communication costs required to synchronize tensor updates across nodes, but the exact impact will vary based on specific infrastructure and networking capabilities (e.g., whether InfiniBand is used).

Table 6: Wall-clock time comparison between Sst and random sampling on different model sizes. Although Sst introduces evaluation overhead, the overall time reduction remains substantial because of the smaller training subset (30k vs. 100k). The values are averaged over 3 runs, and the standard error is shown.

| | | Time (hrs) | | |
|---|---|---|---|---|
| Method | Num GPUs | Random 100k | Sst | Sst Overhead |
| Q2.5-0.5B | 8 | $2.15 \pm 0.05$ | $1.20 \pm 0.07$ | 40% |
| L3.1-8B | 8 | $2.70 \pm 0.06$ | $2.10 \pm 0.04$ | 60% |
| Qwen2.5-32B | 8 | $36.70 \pm 0.01$ | $16.80 \pm 0.03$ | 34% |

## C.2 Optimized Inference

This sections describes the experimental setup used to analyze how inference-optimized backends, such as vLLM, can reduce the overhead introduced by Sst, as presented in §4.2. The experiments follow the same setup detailed in §3. However, we restrict the models to the ones from the Qwen 2.5 family to ensure a consistent comparison across different settings. To evaluate the improvement, we track any additional overhead introduced by Sst during training. To use vLLM (Kwon et al., 2023), we implemented a custom training loop based on (Lambert et al., 2024) where, we pause training, save the training state (model, optimizer, dataloader, etc.), unload the model and optimizer to free GPU memory, and then evaluate the 100k dataset with vLLM. Once evaluation is complete, the training state is reloaded to resume training. This approach introduces overhead from saving and loading the trainer state that is non-negligible for large model states, but it is used solely for this analysis and not in the main experiments in §5. We evaluated vLLM with Bfloat16, 8-bit, and 4-bit precision, using the results to Bfloat16 without vLLM as comparison reference. The findings are presented in Figure 6.

# D Detailed Analysis on Static Perplexity Sampling

We start by introducing results on the effectiveness and limitations of static perplexity-based data selection in IFT. These findings motivate and lay the groundwork for our proposed adaptive method, introduced in §4.

The work of Marion et al. (2023) demonstrates that simple perplexity-based data selection outperforms more complex metrics. However, that analysis is limited to pre-training, with no comment on its broader applicability to downstream tasks. Additionally, their study was limited to two models (124M and 1.5B parameters), leaving open questions about the generality of these findings across different model sizes and architectures (e.g., Llama vs. Gemma). Building on their methodology, we extend the analysis with several key distinctions: **IFT Setting**: We investigate perplexity-based data selection in the context of IFT rather than pre-training. **Target Model as Reference**: We don't rely on external reference models and use the target model to guide the selection of its training data. **Broad Analysis Scope**: We evaluated models ranging from 0.5B to 32B parameters across three state-of-the-art architectures, offering a significantly broader evaluation than the two models used in Marion et al. (2023). Through this analysis, we aim to address the following key questions:

- **Performance and Consistency**: Does perplexity-based data selection perform well in the IFT setting, and how does its effectiveness vary across model sizes and architectures?
- **Impact of Training on Selection Performance**: Does the performance of perplexity-based data selection improve with training?
- **Selection Criteria Across Training Stages**: Is the criteria for selecting data based on perplexity consistent throughout training, or does it need to be adapted to achieve consistent performance?

### D.1 Experimental Setup

We conduct experiments using a stratified 100k subsample of the Tulu 3 SFT Mix dataset (Lambert et al., 2024), which spans 15 diverse and recent datasets (e.g., No Robots (Rajani et al., 2023), Aya (Singh et al., 2024), NuminaMath-TIR (LI et al., 2024)). We chose this subsample to allow for rigorous experimentation given the computational resources available, while ensuring it represents the full mixture, informed by the sampling analysis in (Lambert et al., 2024). We use models from different architectures (Llama3.1 (Grattafiori et al., 2024a), Llama3.2 (Grattafiori et al., 2024a), Qwen2.5 (Yang et al., 2024), and Gemma2 (Team et al., 2024)), ranging from 0.5B to 32B parameters to better understand the impact of different model characteristics (e.g., pre-training data composition, size) on perplexity-based data selection performance. We use full-parameter training for two epochs using the setup proposed by (Lambert et al., 2024). We assess performance on the newer Open LLM Leaderboard 2 (Fourrier et al., 2024), which includes more challenging benchmarks compared to the earlier version (Beeching et al., 2023), including IFEval (Zhou et al., 2023b), BBH (Suzgun et al., 2022), MATH LvL 5 (Hendrycks et al., 2021), GPQA (Rein et al., 2023), MuSR (Sprague et al., 2024), and MMLU-PRO (Wang et al., 2024). We provide detailed data, training, evaluation, and data selection setups in Appendix A, and in subsequent sections, provide the methods and baselines used to address each key questions outlined above.

### D.2 Performance and Consistency

This section investigates the effectiveness of static perplexity-based data selection for IFT across diverse models architectures and sizes. The primary objective is to assess the effectiveness of static perplexity-based selection for IFT and to analyze how its impact changes with model size and pre-training characteristics.

**Methods:** For this analysis we consider four data selection strategies. Random: from (1) the full 100k mixture or (2) per-dataset (Random, Random Per-Dataset). Static perplexity-based: from the bottom, middle, and top segments of (3) the overall mixture (Keep Bottom, Keep Middle, Keep Top) or (4) per-dataset (Keep Bottom Per-Dataset, Keep Middle Per-Dataset, Keep Top Per-Dataset). For each method, we select subsets of 10%, 30%, and 50% of the full 100k mixture, following (Marion et al., 2023). Given the large-scale nature of our experiments (248 training runs), we limit the evaluation of random selection strategies to two independent runs with different seeds (123 and 42)—shown to be effective by (Lambert et al., 2024). This maintains computational practicality while still capturing some measure of variance. Each method is compared against the full 100k data mixture baseline using its average score across all benchmarks (§D.1). **Baseline:** We contrast the above strategies against a baseline that uses the full 100k data mixture (100%). **Models:** To ensure the generalizability of our findings, we evaluate a diverse set of models varying in size, pre-training data composition, and architectural design. Specifically, we use Qwen2.5 0.5B, Llama3.2 1B, Llama3.2 3B, Llama3.1 8B, Gemma2 9B, Qwen2.5 14B, Gemma2 27B, and Qwen2.5 32B. We selected these models to study the impact of unique model characteristics on static perplexity-based data selection performance. In total, we trained 8 models, each with 31 different training configurations, resulting in **248 training runs for this experiment alone**.

Figure 2 shows the strongest-performing baseline, Random Per-Dataset with the top-performing static perplexity-based strategies: Keep Bottom Per-Dataset, Keep Middle Per-Dataset, and Keep Top Per-Dataset. Full results, including all methods and baselines, and per-benchmark results, are detailed in Appendix F. The relative performance change $\Delta R$ is calculated as:

$$\Delta R = \frac{S_{\text{method}} - S_{\text{baseline}}}{S_{\text{baseline}}} \tag{7}$$

where $S_{\text{method}}$ and $S_{\text{baseline}}$ are the average Open LLM Leaderboard score of the method's and the baseline respectively. This metric allows consistent evaluation across models with varying baseline performance. These results highlight several key insights:

**Heuristic-based data selection is insufficient for consistent performance.** Some models, even with naive random selection, outperform the 100% baseline, indicating that while the extensive heuristic and empirical-based approach of Lambert et al. (2024) is effective, further refinements could yield even greater performance gains simply by optimizing training resource allocation. It also suggests opportunities

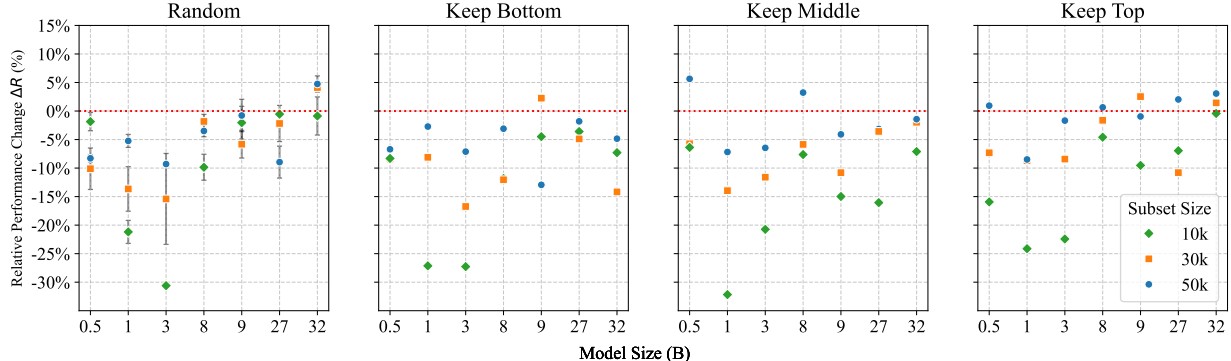

Figure 8: Performance of static perplexity sampling compared to random selection from all the mixutre. Relative performance change $\Delta R$ is calculated as $\Delta R = (S_{\mathrm{method}} - S_{\mathrm{baseline}})/S_{\mathrm{baseline}}$, where $S_{\mathrm{method}}$ and $S_{\mathrm{baseline}}$ are the average Open LLM Leaderboard score of the method's and 100% baseline. Points below the red dashed line indicate performance drops compared to the baseline, and error bars show the standard error over two random seeds.

to enhance computational efficiency, reduce training costs, and improve scalability for larger models.

**Dataset-aware selection produces the best results.** Per-dataset selection consistently outperforms full-mix sampling across all models for both the random baseline and perplexity-based strategies. This effect is particularly evident in varied-complexity datasets like ours, where FLAN v2 examples are easier than more reasoning-intensive tasks such as NuminaMath. This issue is amplified when selecting from the overall mix using a complexity-aware method, like DEITA, INSTAG, or static perplexity-based strategies, as they skew the subset toward harder examples. However, IFT models require exposure to a diverse range datasets or domains, including easier ones, and dataset-aware sampling preserves this balance.

**Static perplexity sampling improves on baseline but lacks consistency.** In Figure 2, we observe that keeping the bottom segment consistently underperforms naive random selection, suggesting that the most useful data lies in the top 50% of the perplexity distribution. This is likely due to data leakage (Lambert et al., 2024), where models may have encountered these examples during pre-training, which could explain why the effect is more pronounced in some models (e.g., Llama3.2 1B and 3B) than others. Another possible explanation is that modern LLMs, like those in our study, are more capable due to training on trillions of tokens (Yang et al., 2024). In both cases, training compute is allocated inefficiently leading to suboptimal performance. In contrast, per-dataset middle-segment selection tends to improve performance for models smaller than 14B parameters, while keeping the top segment benefits larger models, such as, Qwen2.5-32B, which matches the performance of the 100% baseline using only 10% of the data. However, for models smaller than 27B, Random Per-Dataset remains a strong baseline, suggesting that static perplexity sampling alone is insufficient for consistent performance. Moreover, the best-performing configuration varies across models and subset sizes, highlighting the need for an adaptive data selection strategy for robust results.

### D.3   Impact of Training on Selection Performance

The work of Marion et al. (2023) showed that data selection performance improves with better reference models, either larger in size or trained on better data. In this section, we investigate whether similar behavior occurs when using the target model as the reference as it trains on more data. Specifically, we aim to identify when in the training process data selection performance peaks and how it evolves with additional training.

**Methods:** We select three models (Qwen2.5-0.5B, Llama3.1-8B, and Qwen2.5-32B) and use the perplexity sampling configurations from the previous section: Keep Top Per-Dataset for Qwen2.5-32B and Keep Middle Per-Dataset for the other two models. We first train each model on data randomly sampled per dataset, following the best-performing random strategy from the previous section (Random Per-Dataset). At specific points during the two-epoch training process (5%, 10%, 15%, 25%, 50%, and 75%), we compute the perplexity values and apply perplexity sampling to select 30% of the 100k dataset. We then continue training the model

for the remaining iterations to complete two full epochs. For each method, we conduct two runs with different random seeds (123, 42), and report the average score and standard error across the runs. **Baseline:** We use results from the previous experiment, where we selected 30% of the 100k mixture using the pre-trained model as the reference at the start of training. Figure 4 shows the results of this analysis.

**Data selection performance does not always improve with training.** As shown in Figure 4, performance improves when perplexity sampling is delayed to a certain point in training but degrades beyond this threshold, which varies across models we tested and generally decreases with model size. For instance, Qwen2.5-0.5B benefits from delaying selection until 25% of the training process, while Qwen2.5-32B starts to degrade as early as 10%. Llama3.1-8B and Qwen2.5-32B exhibit significant performance drops after the performance peak, with Qwen2.5-32B being particularly affected. Our analysis (Figure 9) attributes this behavior to changes in the perplexity distribution. For Llama3.1-8B, the distribution becomes heavy-tailed as the model trains on more data, which shifts focus toward overly challenging examples. For Qwen2.5-32B, the distribution becomes narrower and skews toward the top segment, emphasizing overly complex (e.g., Table 7) and noisy examples (e.g., Table 8). In contrast, Qwen2.5-0.5B is less influenced by these distributional changes as we show in Figure 9, where the distribution remains relatively stable, explaining why it benefits from a longer delay in perplexity-based selection. We also observe that these changes also aligns approximately with major trends changes in the overall training loss (Figure 9) that we describe in more details in §4. These findings suggest that an adaptive data selection strategy, which delays perplexity-based selection to an optimal point that varies by model, is essential for achieving consistent performance.

### D.4 Selection Criteria Across Training Stages

In this section, we investigate whether using different segments of the perplexity distribution at various training stages can improve the final performance of perplexity-based data selection. The goal is to determine if modulating the selection window is necessary to achieve consistent performance.

**Methods:** For this experiment, we use Llama3.1-8B and Qwen2.5-32B and compare the performance of the best and second-best static perplexity sampling configuration from the previous section. We use Keep Middle and Bottom Per-Dataset for Llama3.1-8B, and Keep Top and Middle Per-Dataset for Qwen2.5-32B. We omit Qwen2.5-0.5B from this analysis, as its performance variation does not show meaningful comparisons.

**The best perplexity distribution segment varies during training.** Figure 3 shows that using different perplexity segments at different training stages affects performance. For example, both Qwen2.5-32B and Llama3.1-8B benefit from starting with easier examples (i.e., using a lower perplexity segment) before transitioning to more challenging examples. When contrasting this behavior in Figure 3 to the performance peaks observed in Figure 4, we find that these transitions align approximately with the performance peaks. Our analysis suggests that using easy examples earlier in training (i.e., lower perplexity segments) stabilizes learning by allowing enough training iterations to adapt to the data distribution shift that occurs early in training before handling more complex examples effectively. This aligns with early research on curriculum learning (Bengio et al., 2009), which demonstrated that starting with easier examples can enhance learning efficiency. These results raise the question of whether an adaptive selection strategy, capable of modulating the selection window dynamically, can offset the initial performance drop and perhaps improve final performance (we show in Table 2 that it does).

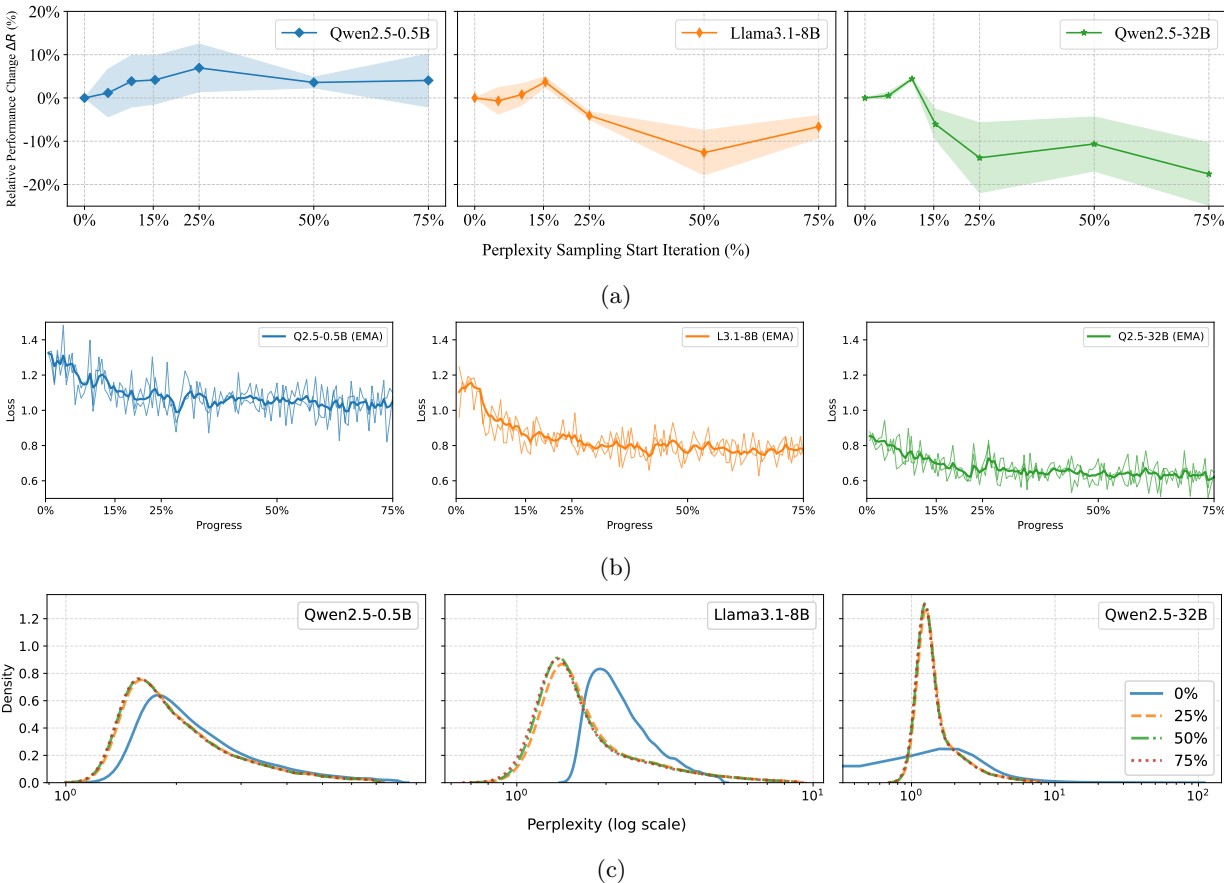

Figure 9: Effect of delaying perplexity sampling on model learning. Performance peaks (a) approximately when the loss begins to flatten (b). Further delay in perplexity sampling degrades performance due to a long-tail perplexity distribution, causing the subset to include disproportionately many complex or noisy examples (high perplexity). (a) Repeats Figure4 for easy comparison of training loss and distribution behaviors. (b) Displays overall training loss over time. (c) Shows perplexity distribution at different training iterations.

# E   Detailed Results

## E.1   Intractable Examples

In this sections, we present examples of intractable examples that represent mislabeled or noisy data that we identified during our analysis.

Table 7: First 50 out of 133 lines from an extremely hard example found during our analysis

| Instruction | Create a program to evaluate a postfix expression. |
|---|---|
| Output | |

```python
# Python program to evaluate value of a postfix expression

# Class to convert the expression
class Conversion:

    # Constructor to initialize the class variables
    def __init__(self, capacity):
        self.top = -1
        self.capacity = capacity
        # This array is used a stack
        self.array = []
        # Precedence setting
        self.output = []
        self.precedence = {'+':1, '-':1, '*':2, '/':2, '^':3}

    # check if the stack is empty
    def isEmpty(self):
        return True if self.top == -1 else False

    # Return the value of the top of the stack
    def peek(self):
        return self.array[-1]

    # Pop the element from the stack
    def pop(self):
        if not self.isEmpty():
            self.top -= 1
            return self.array.pop()
        else:
            return "$"

    # Push the element to the stack
    def push(self, op):
        self.top += 1
        self.array.append(op)

    # A utility function to check is the given character
    # is operand
    def isOperand(self, ch):
        return ch.isalpha()

    # Check if the precedence of operator is strictly
    # less than top of stack or not
    def notGreater(self, i):
        try:
            a = self.precedence
            b = self.precedence
            return True if a  <= b else False
        except KeyError:
            return False
```

Table 8: Ambiguous example found during our analysis. The target output contains place holders (________),
suggesting ChatGPT generated data.

| |
| --- |
| **Inst**: Gen a summary of an article containing more than 500 words. |
| **Out**: The article is about ________. It discusses the subject in depth, exploring various aspects of the topic. It provides a comprehensive overview of different perspectives on the topic and presents evidence to support its conclusions. In conclusion, the article provides a comprehensive evaluation of ________ by offering a balanced and unbiased assessment of the subject. |

Table 9: Performance of static perplexity sampling compared to random selection and 100% baseline for Qwen2.5-0.5B.

| Method | Seed | Subset | ΔR | Avg | MATH Lvl 5 | IFEval | MMLU-PRO | GPQA | MUSR | BBH |
|---|---|---|---|---|---|---|---|---|---|---|
| Random Per-Dataset | 42 | 0.5 | 0.12 | 8.79 | 1.59 | 33.05 | 4.98 | 2.57 | 1.76 | 4.46 |
| Keep Middle | 42 | 0.5 | 0.06 | 8.31 | 1.21 | 30.01 | 4.89 | 4.03 | 1.43 | 3.42 |
| Keep Middle Per-Dataset | 42 | 0.5 | 0.04 | 8.2 | 0.98 | 29.18 | 6.56 | 3.02 | 1.27 | 4.12 |
| Random Per-Dataset | 123 | 0.5 | 0.04 | 8.19 | 1.36 | 30.71 | 7.03 | 0.56 | 1.27 | 4.33 |
| Random Per-Dataset | 123 | 0.3 | 0.02 | 8.05 | 1.59 | 30.15 | 5.35 | 0.89 | 2.25 | 5.17 |
| Keep Top | 42 | 0.5 | 0.01 | 7.94 | 0.98 | 29.61 | 5.32 | 1.79 | 2.02 | 4.71 |
| Baseline | 42 | 1.0 | 0.0 | 7.87 | 0.91 | 31.07 | 5.88 | 0.22 | 1.27 | 4.56 |
| Random | 123 | 0.1 | 0.0 | 7.85 | 1.81 | 26.59 | 7.03 | 1.9 | 1.92 | 3.09 |
| Keep Middle Per-Dataset | 42 | 0.3 | −0.01 | 7.76 | 1.96 | 25.63 | 3.9 | 6.04 | 1.27 | 6.22 |
| Random Per-Dataset | 42 | 0.3 | −0.03 | 7.64 | 0.91 | 29.09 | 6.65 | 0.11 | 1.43 | 4.14 |
| Keep Top Per-Dataset | 42 | 0.5 | −0.03 | 7.64 | 0.91 | 28.86 | 6.55 | 0.45 | 1.43 | 4.6 |
| Random | 42 | 0.1 | −0.03 | 7.6 | 2.04 | 24.94 | 6.33 | 1.9 | 2.78 | 3.28 |
| Random | 123 | 0.3 | −0.06 | 7.36 | 1.28 | 27.03 | 5.03 | 0.56 | 2.9 | 6.04 |
| Keep Middle | 42 | 0.3 | −0.06 | 7.42 | 1.66 | 26.94 | 3.69 | 2.57 | 2.23 | 5.64 |
| Random Per-Dataset | 42 | 0.1 | −0.06 | 7.43 | 1.59 | 26.09 | 6.68 | 1.01 | 1.76 | 4.56 |
| Keep Middle | 42 | 0.1 | −0.06 | 7.37 | 1.66 | 25.06 | 5.29 | 1.68 | 3.15 | 2.96 |
| Keep Top Per-Dataset | 42 | 0.3 | −0.07 | 7.32 | 1.51 | 25.73 | 5.99 | 1.45 | 1.92 | 4.57 |
| Keep Top | 42 | 0.3 | −0.07 | 7.29 | 1.36 | 27.25 | 5.7 | 0.89 | 1.27 | 4.5 |
| Keep Bottom | 42 | 0.5 | −0.07 | 7.34 | 1.36 | 27.46 | 6.78 | 0.0 | 1.11 | 5.62 |
| Random | 123 | 0.5 | −0.08 | 7.27 | 1.44 | 27.54 | 5.66 | 0.0 | 1.7 | 3.82 |
| Keep Bottom | 42 | 0.1 | −0.08 | 7.22 | 2.19 | 24.75 | 6.65 | 0.89 | 1.6 | 3.3 |
| Keep Bottom | 42 | 0.3 | −0.08 | 7.21 | 1.66 | 26.84 | 6.43 | 0.0 | 1.11 | 6.09 |
| Keep Top Per-Dataset | 42 | 0.1 | −0.08 | 7.22 | 2.19 | 25.41 | 5.22 | 1.34 | 1.92 | 3.62 |
| Keep Bottom Per-Dataset | 42 | 0.5 | −0.08 | 7.23 | 0.98 | 28.49 | 5.55 | 0.0 | 1.11 | 5.58 |
| Keep Bottom Per-Dataset | 42 | 0.1 | −0.08 | 7.25 | 1.36 | 26.97 | 6.38 | 0.11 | 1.43 | 3.58 |
| Random | 42 | 0.5 | −0.09 | 7.17 | 1.28 | 27.27 | 4.74 | 1.12 | 1.43 | 4.79 |
| Keep Middle Per-Dataset | 42 | 0.1 | −0.09 | 7.19 | 1.59 | 24.6 | 6.67 | 1.34 | 1.76 | 3.97 |
| Keep Bottom Per-Dataset | 42 | 0.3 | −0.11 | 7.03 | 1.51 | 26.45 | 6.24 | 0.0 | 0.94 | 4.22 |
| Random | 42 | 0.3 | −0.14 | 6.79 | 1.06 | 25.08 | 4.75 | 1.45 | 1.6 | 4.23 |
| Random Per-Dataset | 123 | 0.1 | −0.15 | 6.68 | 1.96 | 24.78 | 5.73 | 0.0 | 0.94 | 3.61 |
| Keep Top | 42 | 0.1 | −0.16 | 6.62 | 0.98 | 25.36 | 4.92 | 0.22 | 1.6 | 3.53 |

Table 10: Performance of static perplexity sampling compared to random selection and 100% baseline for Llama3.2-1B.

| Method | Seed | Subset | ΔR | Avg | MATH Lvl 5 | IFEval | MMLU-PRO | GPQA | MUSR | BBH |
|---|---|---|---|---|---|---|---|---|---|---|
| Random Per-Dataset | 123 | 0.5 | 0.08 | 8.57 | 0.45 | 36.29 | 1.71 | 0.0 | 4.39 | 3.24 |
| Baseline | 42 | 1.0 | 0.0 | 7.94 | 0.45 | 33.96 | 1.66 | 0.0 | 3.62 | 4.01 |
| Random Per-Dataset | 42 | 0.5 | 0.0 | 7.93 | 0.38 | 34.68 | 1.31 | 0.0 | 3.29 | 4.82 |
| Keep Middle Per-Dataset | 42 | 0.5 | 0.0 | 7.92 | 0.39 | 34.18 | 1.49 | 0.0 | 3.52 | 5.01 |
| Keep Bottom | 42 | 0.5 | −0.03 | 7.72 | 0.38 | 32.75 | 1.43 | 0.0 | 4.05 | 4.2 |
| Random | 42 | 0.5 | −0.04 | 7.61 | 0.3 | 30.67 | 1.36 | 0.0 | 5.73 | 3.36 |
| Random Per-Dataset | 123 | 0.3 | −0.04 | 7.65 | 0.3 | 33.5 | 1.45 | 0.0 | 3.0 | 3.13 |
| Keep Bottom Per-Dataset | 42 | 0.5 | −0.05 | 7.55 | 0.3 | 34.14 | 2.15 | 0.0 | 1.15 | 4.72 |
| Random | 123 | 0.5 | −0.06 | 7.43 | 0.08 | 34.51 | 1.68 | 0.0 | 0.89 | 3.3 |
| Keep Middle | 42 | 0.5 | −0.07 | 7.37 | 0.6 | 31.82 | 1.61 | 0.0 | 2.81 | 5.82 |
| Keep Top | 42 | 0.5 | −0.08 | 7.26 | 0.3 | 32.16 | 1.27 | 0.0 | 2.59 | 4.25 |
| Keep Bottom | 42 | 0.3 | −0.08 | 7.29 | 0.68 | 30.06 | 2.43 | 0.0 | 3.3 | 4.45 |
| Keep Top | 42 | 0.3 | −0.09 | 7.25 | 0.53 | 31.67 | 1.22 | 0.0 | 2.81 | 3.54 |
| Keep Middle Per-Dataset | 42 | 0.3 | −0.09 | 7.22 | 0.43 | 30.91 | 1.41 | 0.0 | 3.34 | 3.48 |
| Random Per-Dataset | 42 | 0.3 | −0.1 | 7.13 | 0.3 | 31.2 | 1.14 | 0.0 | 3.0 | 3.12 |
| Random | 123 | 0.3 | −0.1 | 7.16 | 0.45 | 32.62 | 1.44 | 0.0 | 1.31 | 3.34 |
| Keep Bottom Per-Dataset | 42 | 0.3 | −0.11 | 7.04 | 0.53 | 30.28 | 1.93 | 0.0 | 2.45 | 3.82 |
| Keep Middle | 42 | 0.3 | −0.14 | 6.83 | 0.38 | 29.19 | 0.88 | 0.0 | 3.7 | 2.95 |
| Keep Top Per-Dataset | 42 | 0.3 | −0.15 | 6.72 | 0.15 | 29.81 | 1.35 | 0.0 | 2.29 | 3.72 |
| Keep Top Per-Dataset | 42 | 0.5 | −0.16 | 6.69 | 0.23 | 30.81 | 1.23 | 0.0 | 1.2 | 3.44 |
| Random Per-Dataset | 123 | 0.1 | −0.17 | 6.58 | 0.53 | 28.14 | 1.91 | 0.0 | 2.32 | 3.05 |
| Random | 42 | 0.3 | −0.18 | 6.54 | 0.3 | 29.6 | 1.39 | 0.0 | 1.43 | 3.62 |
| Random | 42 | 0.1 | −0.19 | 6.42 | 0.45 | 26.66 | 0.92 | 0.0 | 4.05 | 3.16 |
| Random Per-Dataset | 42 | 0.1 | −0.2 | 6.38 | 0.53 | 27.54 | 1.86 | 0.0 | 1.95 | 2.84 |
| Random | 123 | 0.1 | −0.23 | 6.1 | 0.6 | 26.11 | 1.42 | 0.0 | 2.35 | 2.89 |
| Keep Top | 42 | 0.1 | −0.24 | 6.02 | 0.53 | 25.26 | 1.26 | 0.0 | 3.06 | 2.57 |
| Keep Top Per-Dataset | 42 | 0.1 | −0.24 | 6.07 | 0.6 | 25.29 | 1.38 | 0.0 | 3.09 | 3.44 |
| Keep Bottom Per-Dataset | 42 | 0.1 | −0.25 | 5.93 | 0.6 | 26.2 | 2.11 | 0.0 | 0.72 | 3.97 |
| Keep Bottom | 42 | 0.1 | −0.27 | 5.78 | 0.38 | 24.62 | 1.75 | 0.0 | 2.17 | 2.84 |
| Keep Middle Per-Dataset | 42 | 0.1 | −0.28 | 5.68 | 0.53 | 23.17 | 1.55 | 0.0 | 3.14 | 3.5 |
| Keep Middle | 42 | 0.1 | −0.32 | 5.38 | 0.38 | 22.35 | 1.53 | 0.0 | 2.66 | 3.13 |

# F   Additional Results for Static Perplexity Data Selection

In the section we present additional results for the static perplexity-based data selection experiment described in §3.

Table 11: Performance of static perplexity sampling compared to random selection and 100% baseline for Llama3.2-3B.

| Method | Seed | Subset | $\Delta R$ | Avg | MATH Lvl 5 | IFEval | MMLU-PRO | GPQA | MUSR | BBH |
|---|---|---|---|---|---|---|---|---|---|---|
| Keep Middle Per-Dataset | 42 | 0.5 | 0.02 | 16.36 | 1.59 | 49.13 | 16.26 | 3.58 | 11.24 | 16.2 |
| Baseline | 42 | 1.0 | 0.0 | 16.08 | 1.44 | 49.93 | 15.58 | 2.01 | 11.42 | 16.8 |
| Random Per-Dataset | 123 | 0.3 | −0.01 | 15.86 | 1.59 | 49.68 | 13.99 | 2.13 | 11.93 | 11.38 |
| Random Per-Dataset | 42 | 0.5 | −0.02 | 15.72 | 1.81 | 47.03 | 16.18 | 2.24 | 11.32 | 17.07 |
| Keep Top | 42 | 0.5 | −0.02 | 15.81 | 1.74 | 47.84 | 15.27 | 3.36 | 10.82 | 14.7 |
| Random Per-Dataset | 42 | 0.3 | −0.03 | 15.55 | 1.5 | 49.23 | 13.12 | 2.13 | 11.78 | 11.64 |
| Random Per-Dataset | 123 | 0.5 | −0.03 | 15.6 | 1.51 | 49.95 | 15.91 | 0.78 | 9.83 | 14.98 |
| Keep Middle | 42 | 0.5 | −0.06 | 15.04 | 1.66 | 47.84 | 15.11 | 1.12 | 9.46 | 12.02 |
| Random | 42 | 0.3 | −0.07 | 14.88 | 1.81 | 45.02 | 13.33 | 4.03 | 10.21 | 14.62 |
| Keep Bottom | 42 | 0.5 | −0.07 | 14.93 | 1.44 | 47.7 | 14.44 | 2.8 | 8.28 | 15.6 |
| Keep Top | 42 | 0.3 | −0.08 | 14.72 | 2.27 | 47.76 | 14.94 | 1.79 | 6.85 | 13.94 |
| Keep Bottom Per-Dataset | 42 | 0.5 | −0.08 | 14.8 | 1.66 | 47.94 | 15.87 | 0.45 | 8.1 | 13.53 |
| Random | 42 | 0.5 | −0.09 | 14.62 | 1.89 | 47.3 | 13.31 | 1.12 | 9.47 | 15.52 |
| Keep Top Per-Dataset | 42 | 0.5 | −0.09 | 14.67 | 1.59 | 46.54 | 15.72 | 2.8 | 6.72 | 10.17 |
| Random | 123 | 0.5 | −0.1 | 14.55 | 1.89 | 48.31 | 14.39 | 1.23 | 6.91 | 17.74 |
| Keep Bottom Per-Dataset | 42 | 0.3 | −0.12 | 14.22 | 1.28 | 47.04 | 14.26 | 2.01 | 6.5 | 11.95 |
| Keep Middle | 42 | 0.3 | −0.12 | 14.21 | 1.74 | 46.54 | 10.9 | 1.45 | 10.43 | 13.0 |
| Keep Middle Per-Dataset | 42 | 0.3 | −0.13 | 13.92 | 1.36 | 46.1 | 10.29 | 3.02 | 8.85 | 14.02 |
| Random Per-Dataset | 123 | 0.1 | −0.15 | 13.74 | 0.91 | 44.78 | 11.26 | 3.02 | 8.74 | 10.31 |
| Keep Top Per-Dataset | 42 | 0.3 | −0.16 | 13.54 | 1.59 | 44.41 | 12.01 | 3.58 | 6.12 | 12.49 |
| Keep Bottom | 42 | 0.3 | −0.17 | 13.39 | 1.51 | 42.54 | 12.97 | 2.24 | 7.67 | 12.47 |
| Keep Middle | 42 | 0.1 | −0.21 | 12.74 | 1.51 | 39.98 | 11.4 | 3.13 | 7.69 | 9.61 |
| Random Per-Dataset | 42 | 0.1 | −0.21 | 12.73 | 1.89 | 43.29 | 9.2 | 3.13 | 6.12 | 10.97 |
| Keep Top | 42 | 0.1 | −0.22 | 12.47 | 0.83 | 40.67 | 12.72 | 2.8 | 5.33 | 12.24 |
| Random | 123 | 0.3 | −0.23 | 12.32 | 1.51 | 43.01 | 9.9 | 3.13 | 4.04 | 11.45 |
| Keep Top Per-Dataset | 42 | 0.1 | −0.23 | 12.41 | 2.19 | 38.88 | 12.35 | 2.35 | 6.27 | 12.87 |
| Keep Middle Per-Dataset | 42 | 0.1 | −0.26 | 11.96 | 2.42 | 41.29 | 10.79 | 1.34 | 3.98 | 13.1 |
| Keep Bottom | 42 | 0.1 | −0.27 | 11.69 | 1.28 | 37.64 | 9.15 | 2.91 | 7.48 | 9.67 |
| Random | 123 | 0.1 | −0.3 | 11.22 | 2.04 | 39.02 | 9.56 | 2.24 | 3.22 | 8.03 |
| Keep Bottom Per-Dataset | 42 | 0.1 | −0.3 | 11.32 | 1.36 | 38.07 | 9.49 | 2.35 | 5.31 | 8.77 |
| Random | 42 | 0.1 | −0.31 | 11.09 | 1.44 | 38.65 | 7.56 | 1.9 | 5.92 | 9.53 |

Table 12: Performance of static perplexity sampling compared to random selection and 100% baseline for Llama3.1-8B.

| Method | Seed | Subset | $\Delta R$ | Avg | MATH Lvl 5 | IFEval | MMLU-PRO | GPQA | MUSR | BBH |
|---|---|---|---|---|---|---|---|---|---|---|
| Keep Top Per-Dataset | 42 | 0.3 | 0.03 | 22.72 | 5.06 | 60.74 | 24.52 | 7.38 | 15.91 | 28.61 |
| Keep Middle Per-Dataset | 42 | 0.5 | 0.03 | 22.9 | 6.23 | 60.28 | 23.84 | 4.25 | 14.63 | 28.19 |
| Keep Middle | 42 | 0.5 | 0.03 | 22.86 | 5.59 | 61.29 | 24.36 | 5.15 | 17.91 | 27.95 |
| Keep Middle Per-Dataset | 42 | 0.3 | 0.01 | 22.31 | 5.74 | 60.9 | 23.26 | 5.96 | 15.7 | 29.39 |
| Keep Top | 42 | 0.5 | 0.01 | 22.29 | 4.38 | 62.04 | 22.84 | 4.36 | 17.81 | 25.92 |
| Baseline | 42 | 1.0 | 0.0 | 22.14 | 5.44 | 61.11 | 23.39 | 4.81 | 15.97 | 29.53 |
| Random | 42 | 0.3 | −0.01 | 22.01 | 5.51 | 57.85 | 22.0 | 3.8 | 20.9 | 28.13 |
| Random Per-Dataset | 123 | 0.3 | −0.01 | 21.99 | 5.29 | 61.85 | 21.8 | 5.7 | 15.32 | 16.71 |
| Random Per-Dataset | 42 | 0.5 | −0.01 | 22.03 | 6.19 | 60.78 | 23.08 | 6.6 | 13.48 | 24.25 |
| Keep Bottom Per-Dataset | 42 | 0.5 | −0.02 | 21.62 | 5.36 | 61.28 | 21.9 | 4.59 | 14.99 | 24.52 |
| Random Per-Dataset | 42 | 0.3 | −0.02 | 21.73 | 5.01 | 61.65 | 21.41 | 5.6 | 14.98 | 16.12 |
| Random | 42 | 0.5 | −0.02 | 21.59 | 6.19 | 58.73 | 24.17 | 4.7 | 14.18 | 24.35 |
| Keep Top | 42 | 0.3 | −0.02 | 21.78 | 4.83 | 60.07 | 24.66 | 5.7 | 13.66 | 24.47 |
| Keep Bottom Per-Dataset | 42 | 0.3 | −0.03 | 21.46 | 5.21 | 57.03 | 24.4 | 4.25 | 16.42 | 26.57 |
| Random | 123 | 0.3 | −0.03 | 21.47 | 6.12 | 60.28 | 22.64 | 3.91 | 14.39 | 28.09 |
| Random Per-Dataset | 123 | 0.1 | −0.03 | 21.47 | 6.34 | 56.73 | 22.77 | 5.37 | 16.13 | 24.54 |
| Keep Bottom | 42 | 0.5 | −0.03 | 21.46 | 7.02 | 58.89 | 21.65 | 4.14 | 15.59 | 25.79 |
| Random Per-Dataset | 42 | 0.1 | −0.04 | 21.27 | 4.76 | 54.42 | 24.55 | 6.26 | 16.37 | 23.33 |
| Random | 123 | 0.5 | −0.05 | 21.15 | 5.82 | 58.83 | 21.2 | 5.37 | 14.51 | 27.26 |
| Keep Top | 42 | 0.1 | −0.05 | 21.13 | 5.44 | 51.52 | 24.25 | 6.71 | 17.72 | 20.74 |
| Keep Top Per-Dataset | 42 | 0.5 | −0.05 | 21.05 | 4.38 | 58.28 | 21.04 | 6.38 | 15.16 | 22.7 |
| Keep Top Per-Dataset | 42 | 0.1 | −0.06 | 20.74 | 5.36 | 55.51 | 21.89 | 6.82 | 14.1 | 23.28 |
| Keep Middle | 42 | 0.3 | −0.06 | 20.85 | 4.53 | 53.17 | 25.43 | 6.6 | 14.51 | 27.32 |
| Random Per-Dataset | 123 | 0.5 | −0.07 | 20.58 | 5.44 | 60.53 | 23.03 | 2.46 | 11.45 | 28.98 |
| Random | 42 | 0.1 | −0.08 | 20.47 | 5.29 | 47.58 | 22.89 | 6.15 | 20.42 | 22.53 |
| Keep Middle | 42 | 0.1 | −0.08 | 20.45 | 5.36 | 51.89 | 22.55 | 5.03 | 17.44 | 22.31 |
| Keep Middle Per-Dataset | 42 | 0.1 | −0.11 | 19.74 | 6.5 | 50.77 | 22.15 | 2.24 | 17.02 | 22.84 |
| Random | 123 | 0.1 | −0.12 | 19.46 | 4.46 | 48.21 | 22.77 | 6.6 | 15.25 | 23.25 |
| Keep Bottom | 42 | 0.3 | −0.12 | 19.48 | 6.42 | 51.89 | 20.94 | 5.15 | 12.98 | 26.31 |
| Keep Bottom | 42 | 0.1 | −0.12 | 19.52 | 5.36 | 45.88 | 23.48 | 8.28 | 14.62 | 26.2 |
| Keep Bottom Per-Dataset | 42 | 0.1 | −0.13 | 19.16 | 5.29 | 48.67 | 22.08 | 5.15 | 14.62 | 28.08 |

Table 13: Performance of static perplexity sampling compared to random selection and 100% baseline for Gemma2-9B.

| Method | Seed | Subset | ΔR | Avg | MATH Lvl 5 | IFEval | MMLU-PRO | GPQA | MUSR | BBH |
|---|---|---|---|---|---|---|---|---|---|---|
| Random Per-Dataset | 42 | 0.5 | 0.05 | 29.19 | 12.99 | 66.95 | 29.64 | 10.4 | 25.97 | 35.37 |
| Keep Top | 42 | 0.3 | 0.03 | 28.51 | 10.5 | 69.46 | 33.34 | 11.74 | 17.53 | 36.8 |
| Keep Top Per-Dataset | 42 | 0.3 | 0.03 | 28.55 | 12.16 | 67.95 | 31.06 | 7.83 | 23.75 | 30.59 |
| Random | 123 | 0.5 | 0.02 | 28.38 | 13.07 | 69.47 | 30.41 | 10.4 | 18.56 | 35.93 |
| Keep Middle Per-Dataset | 42 | 0.5 | 0.02 | 28.4 | 11.48 | 66.39 | 30.5 | 10.29 | 23.33 | 33.02 |
| Keep Bottom | 42 | 0.3 | 0.02 | 28.44 | 20.17 | 61.43 | 32.15 | 8.95 | 19.49 | 32.54 |
| Random | 123 | 0.1 | 0.01 | 28.04 | 16.16 | 62.9 | 31.09 | 13.09 | 16.96 | 35.39 |
| Keep Top Per-Dataset | 42 | 0.1 | 0.0 | 27.86 | 13.44 | 61.84 | 35.27 | 6.71 | 22.06 | 34.09 |
| Baseline | 42 | 1.0 | 0.0 | 27.81 | 15.26 | 66.74 | 29.52 | 8.95 | 18.58 | 31.56 |
| Random Per-Dataset | 42 | 0.3 | 0.0 | 27.92 | 11.93 | 65.36 | 30.8 | 9.96 | 21.56 | 33.47 |
| Keep Middle Per-Dataset | 42 | 0.3 | 0.0 | 27.94 | 14.95 | 65.4 | 29.21 | 12.3 | 17.86 | 33.58 |
| Keep Top | 42 | 0.5 | −0.01 | 27.54 | 10.5 | 67.61 | 29.42 | 12.64 | 17.55 | 36.71 |
| Random Per-Dataset | 123 | 0.5 | −0.01 | 27.57 | 14.65 | 68.61 | 29.26 | 7.61 | 17.73 | 32.93 |
| Random Per-Dataset | 123 | 0.3 | −0.02 | 27.28 | 14.2 | 66.39 | 30.33 | 9.28 | 16.2 | 33.74 |
| Random Per-Dataset | 123 | 0.1 | −0.03 | 27.04 | 16.39 | 62.6 | 29.48 | 8.72 | 18.03 | 33.94 |
| Random | 42 | 0.3 | −0.03 | 26.87 | 12.76 | 65.63 | 32.77 | 8.95 | 14.23 | 33.67 |
| Keep Bottom Per-Dataset | 42 | 0.5 | −0.03 | 26.96 | 14.27 | 61.74 | 31.01 | 9.96 | 17.84 | 35.12 |
| Keep Bottom Per-Dataset | 42 | 0.3 | −0.03 | 27.05 | 16.69 | 64.97 | 29.59 | 8.28 | 15.73 | 32.41 |
| Random | 42 | 0.5 | −0.04 | 26.8 | 13.29 | 69.04 | 31.23 | 8.61 | 11.85 | 33.59 |
| Random Per-Dataset | 42 | 0.1 | −0.04 | 26.58 | 15.48 | 58.72 | 26.05 | 7.72 | 24.92 | 32.42 |
| Keep Bottom | 42 | 0.1 | −0.04 | 26.56 | 18.43 | 53.02 | 30.53 | 12.08 | 18.75 | 35.51 |
| Keep Bottom Per-Dataset | 42 | 0.1 | −0.04 | 26.82 | 15.63 | 59.04 | 33.8 | 12.08 | 13.54 | 36.11 |
| Keep Middle | 42 | 0.5 | −0.04 | 26.67 | 12.54 | 66.54 | 27.96 | 9.51 | 16.81 | 34.97 |
| Random | 42 | 0.1 | −0.05 | 26.44 | 15.71 | 59.67 | 30.88 | 9.62 | 16.33 | 32.78 |
| Keep Middle Per-Dataset | 42 | 0.1 | −0.06 | 26.24 | 12.92 | 60.86 | 31.76 | 8.5 | 17.18 | 33.43 |
| Keep Top Per-Dataset | 42 | 0.5 | −0.07 | 25.77 | 10.8 | 68.19 | 26.12 | 7.49 | 16.25 | 31.76 |
| Random | 123 | 0.3 | −0.08 | 25.52 | 12.54 | 64.38 | 26.71 | 9.28 | 14.68 | 32.33 |
| Keep Top | 42 | 0.1 | −0.1 | 25.16 | 8.84 | 56.32 | 33.72 | 11.74 | 15.18 | 36.06 |
| Keep Middle | 42 | 0.3 | −0.11 | 24.81 | 11.1 | 60.56 | 29.01 | 11.52 | 11.84 | 33.46 |
| Keep Bottom | 42 | 0.5 | −0.13 | 24.21 | 17.15 | 57.12 | 28.12 | 4.7 | 13.97 | 33.45 |
| Keep Middle | 42 | 0.1 | −0.15 | 23.64 | 9.52 | 55.29 | 30.86 | 9.4 | 13.15 | 34.31 |

Table 14: Performance of static perplexity sampling compared to random selection and 100% baseline for Gemma2-9B.

| Method | Seed | Subset | ΔR | Avg | MATH Lvl 5 | IFEval | MMLU-PRO | GPQA | MUSR | BBH |
|---|---|---|---|---|---|---|---|---|---|---|
| Keep Top Per-Dataset | 42 | 0.5 | 0.01 | 36.14 | 25.72 | 67.93 | 45.38 | 17.44 | 24.23 | 45.31 |
| Baseline | 42 | 1.0 | 0.0 | 35.68 | 24.06 | 70.41 | 45.93 | 14.54 | 23.48 | 42.29 |
| Random Per-Dataset | 42 | 0.5 | 0.0 | 35.63 | 24.55 | 68.65 | 44.65 | 16.11 | 24.19 | 42.3 |
| Keep Top Per-Dataset | 42 | 0.3 | −0.02 | 34.91 | 23.22 | 67.59 | 44.49 | 17.53 | 21.72 | 42.81 |
| Random Per-Dataset | 123 | 0.5 | −0.03 | 34.6 | 25.6 | 66.98 | 46.26 | 16.0 | 18.16 | 44.39 |
| Random Per-Dataset | 123 | 0.3 | −0.03 | 34.75 | 23.79 | 67.03 | 44.64 | 15.1 | 23.21 | 43.58 |
| Random Per-Dataset | 42 | 0.3 | −0.04 | 34.43 | 20.32 | 67.43 | 45.43 | 17.0 | 21.96 | 44.67 |
| Random | 42 | 0.1 | −0.05 | 34.07 | 29.38 | 60.81 | 42.7 | 15.66 | 21.79 | 42.36 |
| Random | 42 | 0.3 | −0.05 | 33.86 | 25.15 | 65.18 | 46.4 | 16.33 | 16.23 | 42.33 |
| Random | 123 | 0.5 | −0.05 | 33.96 | 25.0 | 68.13 | 43.81 | 15.21 | 17.67 | 40.42 |
| Random | 123 | 0.1 | −0.06 | 33.41 | 29.76 | 60.47 | 42.33 | 13.87 | 20.6 | 35.24 |
| Random | 42 | 0.5 | −0.06 | 33.56 | 26.51 | 64.4 | 44.76 | 15.55 | 16.6 | 45.36 |
| Random Per-Dataset | 123 | 0.1 | −0.07 | 33.18 | 23.19 | 63.89 | 43.73 | 14.21 | 20.9 | 38.05 |
| Keep Top Per-Dataset | 42 | 0.1 | −0.07 | 33.36 | 28.1 | 65.77 | 41.82 | 15.55 | 15.57 | 40.52 |
| Random Per-Dataset | 42 | 0.1 | −0.08 | 32.96 | 20.47 | 65.42 | 43.53 | 14.09 | 21.3 | 38.76 |
| Random | 123 | 0.3 | −0.08 | 32.95 | 23.87 | 64.6 | 45.11 | 15.1 | 16.05 | 43.24 |
| Keep Middle Per-Dataset | 42 | 0.1 | −0.08 | 32.98 | 28.02 | 58.48 | 44.53 | 17.56 | 16.29 | 41.78 |
| Keep Bottom Per-Dataset | 42 | 0.5 | −0.08 | 32.93 | 28.4 | 63.81 | 44.4 | 17.0 | 11.06 | 42.45 |
| Keep Bottom Per-Dataset | 42 | 0.3 | −0.1 | 32.29 | 28.4 | 62.78 | 44.98 | 15.1 | 10.19 | 39.77 |
| Keep Middle Per-Dataset | 42 | 0.5 | −0.1 | 31.95 | 22.66 | 65.39 | 42.67 | 14.88 | 14.14 | 41.35 |
| Keep Middle Per-Dataset | 42 | 0.3 | −0.11 | 31.72 | 27.04 | 62.74 | 41.96 | 14.21 | 12.64 | 42.23 |
| Keep Bottom Per-Dataset | 42 | 0.1 | −0.11 | 31.68 | 24.85 | 63.05 | 45.36 | 14.77 | 10.38 | 36.06 |

Table 15: Performance of static perplexity sampling compared to random selection and 100% baseline for Gemma2-27B.

| Method | Seed | Subset | $\Delta R$ | Avg | MATH Lvl 5 | IFEval | MMLU-PRO | GPQA | MUSR | BBH |
|---|---|---|---|---|---|---|---|---|---|---|
| Keep Top Per-Dataset | 42 | 0.5 | 0.08 | 32.93 | 19.92 | 70.28 | 34.28 | 12.64 | 20.66 | 39.78 |
| Random Per-Dataset | 123 | 0.3 | 0.06 | 32.4 | 20.98 | 68.37 | 38.56 | 10.96 | 21.12 | 41.23 |
| Random Per-Dataset | 42 | 0.3 | 0.04 | 31.59 | 20.0 | 68.79 | 37.86 | 10.0 | 21.32 | 41.3 |
| Keep Top Per-Dataset | 42 | 0.3 | 0.03 | 31.27 | 20.97 | 69.7 | 33.58 | 12.21 | 19.89 | 39.55 |
| Keep Top | 42 | 0.5 | 0.02 | 31.09 | 16.47 | 71.53 | 33.99 | 11.41 | 22.06 | 42.13 |
| Random Per-Dataset | 123 | 0.5 | 0.02 | 30.95 | 22.43 | 72.51 | 30.89 | 9.51 | 19.43 | 38.11 |
| Random Per-Dataset | 123 | 0.1 | 0.01 | 30.69 | 20.62 | 67.58 | 35.16 | 11.19 | 18.92 | 38.98 |
| Random | 42 | 0.3 | 0.01 | 30.77 | 22.73 | 68.39 | 35.63 | 11.41 | 15.71 | 38.23 |
| Keep Top Per-Dataset | 42 | 0.1 | 0.0 | 30.51 | 21.3 | 66.3 | 32.41 | 11.52 | 21.0 | 40.05 |
| Baseline | 42 | 1.0 | 0.0 | 30.47 | 21.68 | 70.26 | 32.18 | 11.41 | 16.84 | 39.68 |
| Keep Middle Per-Dataset | 42 | 0.3 | −0.01 | 30.05 | 19.18 | 67.85 | 33.64 | 12.53 | 17.06 | 33.95 |
| Random | 123 | 0.1 | −0.01 | 30.32 | 20.47 | 69.26 | 32.19 | 12.75 | 16.93 | 38.83 |
| Keep Bottom Per-Dataset | 42 | 0.3 | −0.01 | 30.26 | 22.51 | 71.04 | 33.32 | 13.53 | 10.92 | 40.86 |
| Random | 42 | 0.1 | −0.01 | 30.29 | 21.71 | 68.31 | 32.0 | 12.24 | 17.21 | 39.1 |
| Keep Bottom | 42 | 0.5 | −0.02 | 29.92 | 23.87 | 67.55 | 31.77 | 8.72 | 17.7 | 41.13 |
| Keep Bottom Per-Dataset | 42 | 0.5 | −0.03 | 29.52 | 20.69 | 70.68 | 31.67 | 11.97 | 12.58 | 37.18 |
| Keep Middle | 42 | 0.5 | −0.03 | 29.51 | 17.07 | 73.38 | 31.76 | 11.41 | 13.95 | 38.52 |
| Random Per-Dataset | 42 | 0.1 | −0.04 | 29.27 | 20.62 | 61.98 | 32.97 | 12.3 | 18.47 | 38.39 |
| Keep Bottom | 42 | 0.1 | −0.04 | 29.38 | 24.85 | 55.32 | 37.19 | 12.19 | 17.34 | 40.44 |
| Keep Middle | 42 | 0.3 | −0.04 | 29.38 | 19.79 | 73.15 | 31.21 | 8.72 | 14.05 | 40.48 |
| Random | 123 | 0.3 | −0.05 | 28.84 | 19.49 | 70.92 | 28.08 | 11.74 | 13.98 | 35.83 |
| Keep Bottom | 42 | 0.3 | −0.05 | 28.98 | 22.89 | 63.5 | 35.45 | 12.98 | 10.1 | 43.08 |
| Random | 42 | 0.5 | −0.06 | 28.6 | 20.85 | 68.77 | 30.55 | 10.63 | 12.2 | 33.84 |
| Keep Bottom Per-Dataset | 42 | 0.1 | −0.07 | 28.34 | 21.22 | 64.38 | 34.54 | 11.07 | 10.5 | 42.87 |
| Keep Top | 42 | 0.1 | −0.07 | 28.36 | 11.71 | 60.81 | 36.69 | 12.19 | 20.38 | 41.0 |
| Keep Middle Per-Dataset | 42 | 0.5 | −0.08 | 28.16 | 19.86 | 67.0 | 27.72 | 13.65 | 12.59 | 31.19 |
| Keep Middle Per-Dataset | 42 | 0.1 | −0.09 | 27.77 | 19.18 | 62.05 | 29.69 | 13.31 | 14.6 | 38.6 |
| Keep Top | 42 | 0.3 | −0.11 | 27.18 | 12.24 | 66.64 | 28.82 | 9.4 | 18.82 | 33.47 |
| Random | 123 | 0.5 | −0.12 | 26.89 | 21.07 | 65.7 | 25.35 | 9.28 | 13.06 | 32.2 |
| Random Per-Dataset | 42 | 0.5 | −0.14 | 26.31 | 19.86 | 67.56 | 27.21 | 4.47 | 12.47 | 32.08 |
| Keep Middle | 42 | 0.1 | −0.16 | 25.58 | 12.08 | 63.54 | 27.77 | 7.61 | 16.9 | 38.89 |

Table 16: Performance of static perplexity sampling compared to random selection and 100% baseline for Qwen2.5-32B.

| Method | Seed | Subset | $\Delta R$ | Avg | MATH Lvl 5 | IFEval | MMLU-PRO | GPQA | MUSR | BBH |
|---|---|---|---|---|---|---|---|---|---|---|
| Keep Top Per-Dataset | 42 | 0.5 | 0.1 | 41.53 | 36.1 | 75.03 | 53.25 | 18.46 | 24.79 | 48.85 |
| Random Per-Dataset | 42 | 0.3 | 0.07 | 40.37 | 35.27 | 74.67 | 51.82 | 20.47 | 19.63 | 50.29 |
| Random | 42 | 0.5 | 0.06 | 40.22 | 36.1 | 74.67 | 51.86 | 18.12 | 20.37 | 52.74 |
| Random | 123 | 0.3 | 0.05 | 39.64 | 35.27 | 72.3 | 50.35 | 17.9 | 22.36 | 49.55 |
| Random Per-Dataset | 123 | 0.3 | 0.05 | 39.75 | 36.56 | 74.58 | 48.11 | 18.12 | 21.38 | 49.13 |
| Random Per-Dataset | 42 | 0.5 | 0.04 | 39.45 | 33.69 | 73.42 | 49.76 | 18.57 | 21.8 | 47.12 |
| Random | 42 | 0.3 | 0.04 | 39.26 | 33.91 | 71.25 | 52.15 | 19.02 | 19.98 | 50.24 |
| Random | 123 | 0.5 | 0.03 | 39.15 | 33.91 | 74.39 | 51.45 | 18.23 | 17.78 | 45.45 |
| Keep Top | 42 | 0.5 | 0.03 | 39.05 | 33.16 | 74.03 | 48.77 | 19.69 | 19.61 | 49.87 |
| Random Per-Dataset | 123 | 0.1 | 0.03 | 39.07 | 32.25 | 70.23 | 49.81 | 17.23 | 25.84 | 44.37 |
| Random Per-Dataset | 123 | 0.5 | 0.03 | 38.95 | 35.57 | 75.61 | 46.2 | 16.22 | 21.17 | 49.33 |
| Random | 42 | 0.1 | 0.02 | 38.83 | 35.05 | 68.49 | 47.65 | 17.9 | 25.04 | 46.85 |
| Keep Middle Per-Dataset | 42 | 0.5 | 0.02 | 38.6 | 33.38 | 71.68 | 49.81 | 18.46 | 19.69 | 50.09 |
| Keep Top Per-Dataset | 42 | 0.3 | 0.02 | 38.78 | 33.61 | 73.23 | 51.32 | 19.02 | 16.71 | 49.68 |
| Random Per-Dataset | 42 | 0.1 | 0.02 | 38.51 | 31.8 | 72.53 | 49.81 | 16.33 | 22.06 | 46.09 |
| Keep Top Per-Dataset | 42 | 0.1 | 0.01 | 38.32 | 34.52 | 67.64 | 50.59 | 18.57 | 20.28 | 44.01 |
| Keep Top | 42 | 0.3 | 0.01 | 38.43 | 32.18 | 73.06 | 51.77 | 17.45 | 17.69 | 49.51 |
| Keep Top | 42 | 0.1 | 0.0 | 37.73 | 30.29 | 66.01 | 51.61 | 17.79 | 22.94 | 49.85 |
| Baseline | 42 | 1.0 | 0.0 | 37.89 | 34.67 | 73.9 | 48.03 | 16.33 | 16.52 | 48.06 |
| Keep Middle | 42 | 0.5 | −0.01 | 37.35 | 30.89 | 75.58 | 46.67 | 17.79 | 15.81 | 46.16 |
| Keep Middle Per-Dataset | 42 | 0.3 | −0.02 | 37.04 | 33.99 | 69.01 | 47.66 | 18.46 | 16.09 | 45.72 |
| Keep Middle | 42 | 0.3 | −0.02 | 37.14 | 31.34 | 73.57 | 46.94 | 17.56 | 16.31 | 44.67 |
| Keep Middle Per-Dataset | 42 | 0.1 | −0.03 | 36.58 | 31.42 | 64.87 | 51.99 | 17.11 | 17.52 | 45.36 |
| Keep Bottom Per-Dataset | 42 | 0.5 | −0.03 | 36.94 | 33.84 | 69.37 | 47.47 | 16.44 | 17.57 | 46.6 |
| Random | 123 | 0.1 | −0.04 | 36.29 | 34.44 | 63.2 | 48.88 | 15.55 | 19.4 | 42.08 |
| Keep Bottom Per-Dataset | 42 | 0.3 | −0.04 | 36.44 | 32.33 | 68.28 | 47.44 | 17.67 | 16.46 | 46.5 |
| Keep Bottom Per-Dataset | 42 | 0.1 | −0.05 | 36.05 | 31.19 | 65.23 | 50.2 | 17.34 | 16.27 | 46.05 |
| Keep Bottom | 42 | 0.5 | −0.05 | 36.05 | 29.46 | 67.3 | 46.39 | 16.78 | 20.34 | 45.78 |
| Keep Middle | 42 | 0.1 | −0.07 | 35.2 | 31.5 | 65.23 | 45.62 | 17.23 | 16.41 | 40.25 |
| Keep Bottom | 42 | 0.1 | −0.07 | 35.12 | 25.23 | 61.92 | 49.09 | 17.67 | 21.71 | 45.69 |
| Keep Bottom | 42 | 0.3 | −0.14 | 32.52 | 15.94 | 64.65 | 46.37 | 16.67 | 18.96 | 45.24 |

