# OpenReview forum: "Spaced Scheduling for Large Language Model Training"
_TMLR — Accepted by TMLR_

### Review · Reviewer_eqf2 · 2025-04-05

**Summary Of Contributions:**

This paper proposes a dynamic data selection strategy for training datasets. The core motivation is that aligning data complexity with the model’s training progress can lead to improved performance. Preliminary experiments with a PPL-based heuristic search method reveal that its effectiveness is influenced by both model size and the chosen data subset. To address this, the authors introduce a method that dynamically adjusts the PPL selection range based on feedback from the training loss and the dataset mixing ratio. They also analyze the computational overhead introduced by this selection strategy. Experimental results demonstrate that the proposed method significantly outperforms baseline approaches and even surpasses the performance achieved using the full dataset.

**Audience:**

Yes

**Claims And Evidence:**

Yes

**Requested Changes:**

1. Please add one space after the method name. Currently, the method name and the following words are stacked together. Also, please check the paper writing carefully, there are a lot of typo mistakes like: `but incorporating incorporates`
2. Please rearrange the structure of the paper for the figures. Reading this paper requires frequent back-and-forth referencing of tables and figures. For example, Figure 2 is never mentioned anywhere in the main paper. Some words below Figure 2 refer to Figure 6, which appears only in the appendix. Additionally, some notations are not rigorously defined—for instance, $P_{\text{ref}}^d$ lacks a clear explanation.

**Strengths And Weaknesses:**

Strengths:
1. A novel method for dataset selection for the instruction tuning stage, and the motivation for the method is clear, and the proposed method is sound.
2. The experimental results demonstrate the effectiveness of the proposed method. It has a large improvement compared to the baseline method. The method is evaluated under 7 models with different model sizes


Weaknesses:
1. Although the author begins with the motivation that heuristic methods for data selection are unreliable, the proposed method itself is also heuristic in nature. A theoretical analysis of the data selection strategy, rather than just the computational overhead, would be preferable and would help strengthen the justification for the approach.
2. It's a little bit surprising that the proposed method can even largely surpass the performance of the full dataset. If we disregard the presence of erroneous data in the dataset, the experimental results become somewhat unconvincing that a subset can even surpass the full set. I checked the open LLM leaderboard with the results. For example, for Llama-3.1-8B, the performance for GPQA and MMLU-Pro is 8.05% and 25.42%, which are both higher than the finetuned SST results in this paper (4.25%, 23.84%). I'm not sure if I interpreted correctly with your results, please point it out if I misunderstood something.


Question:

For Section 4.1 Adaptive Scheduling Phase, compute the examples perplexity values PPL(e) (Equation 1) once: Which model is used to compute the ppl? Is this the untrained one or the trained model after warmup? Would the selection / the stage of the model impact the performance, since it directly impacts the distribution of the PPL for that dataset?

---

> ### Author Response · Authors · 2025-04-21
>
> ### Weakness 1
> We thank the reviewer for this valuable comment. We concur that SST incorporates heuristic elements in its adaptive scheduling mechanism.
> SST fundamentally differs from traditional data selection heuristics that are often static and require significant manual tuning as in Tulu 3 (Fig. 3).
> We viewed SST from the theoretical framework of RHO-LOSS work, which we would happily include in the final version.
>
> We see SST as a computationally–cheap approximation to Reducible Hold‑Out Loss Selection (RHO‑LOSS).
>
> $\mathrm{RL}_t(e)=L_t(e)-\mathrm{IL}(e)$: **Reducible loss** (Eq. 3 in RHO‑LOSS)
>
> * **RHO‑LOSS** selects the top‑$k$ points in a pre‑sampled batch by maximizing $\mathrm{RL}_t$.
> * **SST** keeps, for each dataset $d$, the examples whose perplexities lie in a percentile window
>    $[P_{d,t}^{\text{low}},\,P_{d,t}^{\text{high}}]$ centred on $P_{d,t}^{\text{ref}}$ and sized by $\beta_d$.
>
>
>
> **Hypotheses:**
>
> - **H1 (Moderate IL dispersion).**  Within SST’s current window the variance of IL is small enough that $\text{sign}(\mathrm{RL}_t)$ mostly matches $\text{sign}(L_t-c_d)$ where $c_d$ is a small constant.
> - **H2 (Noise‑penalized extremes).**  High‑loss extremes tend to have negative RL (noisy / OOD); very‑low‑loss extremes have RL≈0.
> - **H3 (Slow perceptual drift).**  PPL ranks do not reshuffle drastically between two consecutive window updates.
>
>
> *If* H1–H3 hold at time $t$:
>
> * SST’s retained set $S_t$ is **enriched** in examples whose reducible loss is non‑negative, because
>   – mid‑band points satisfy $L_t\! -\! c_d \approx 0$;
>   – noisy high‑loss points and already‑learned low‑loss points are both outside the window.
> * Most discarded examples are less promising for generalization (expected RL ≤ 0).
> * Thus SST behaves as an **empirical filter** that imitates maximizing RL **without computing RL explicitly**.
>
> ### Weakness 2
>
> 1. Subset Performance vs. Full Dataset:
>
>     The potential for SST to outperform the full dataset, even beyond filtering obviously erroneous data, stems from several factors:
>
>     - Data Efficiency: Redundant examples. The Tulu 3 work found that subsets (e.g., 25-75%) often performed comparably to the full set (Tulu 3 Figure 4 [1]).
>
>     - Adaptation to Pre-training: A static mix might be suboptimal if it overemphasizes topics the model already learned extensively during pre-training
>
>     - Example Complexity: While valid, examples might be intractable for certain models. Eg, complex reasoning tasks (e.g., NuminaMath) might be ineffective or even detrimental for smaller models.
>
>     - Out-of-Distribution Data: Examples may be OOD. Eg, if a model wasn't pre-trained on certain low-resource languages (like the Malayalam example in Table 4).
>
> 2. Llama-3.1-8B Leaderboard Scores:
>
>    - This observation aligns with the known phenomenon of knowledge degradation or "catastrophic forgetting", particularly on knowledge-intensive benchmarks like MMLU-PRO and GPQA ([2-3]). SFT can inadvertently overwrite pre-trained knowledge if the SFT data doesn't sufficiently and explicitly reinforce it.
>         - Supporting this, the Tulu 3 observed performance degradation on knowledge tasks like TruthfulQA after using 5% of the mixture (Figure 4, Table 32 Tulu 3).
>
>     - This degradation is not unique to SST as it is also observed with other methods, including random sampling. Further, this effect may be pronounced in our setup as our data pool did not specifically include data designed to reinforce this knowledge.
>
> It is crucial, however, to consider the overall impact:  Llama-3.1-8B with SST (22.45, Table 1) outperforms all baselines and the leaderboard's performance.
>
> [1]: Tulu 3 https://arxiv.org/abs/2411.15124
> [2] https://openreview.net/forum?id=ScI7IlKGdI
> [3] https://arxiv.org/abs/2406.12227
>
> ### Question
> 1. Model Used for PPL Computation:
>
>    - The individual example scores are computed using the model's state immediately after the warm-up phase.
>    - SST updates the PPL values using the model's state at each iteration from the batch loss values to compute $\beta_d$ and not to rank the examples.
>
> 2. Impact of Model Stage on PPL and Performance:
>
>     - The stage of the model significantly impacts the resulting PPL distribution and subsequent selection performance. We found that computing the PPL values when the loss stabilized provides the best performance as we show in Fig. 4 (delay effect) and 9 (tail behavior).
>
> We updated the manuscript to clearly explain the above.
>
> ### Request Change 1
> We have corrected the spacing issues you noted and thoroughly proofread the manuscript.
>
> ### Request Change 2
> We have revised the manuscript to improve readability based on your feedback regarding the Figure placement, clarify the notations (eg, $P^{ref}_d$), and addressed the unclear mention of Fig. 2.

---

### Review · Reviewer_TeJJ · 2025-04-06

**Summary Of Contributions:**

The paper investigates the perplexity-based sampling method in IFT on LLMs. First, numerical experiments with different sampling strategies are conducted using a static sampling method. The paper argues that for various models, keeping samples of different percentiles is required, and changing the sampling percentile is more beneficial. Then, the paper proposed an adaptive sampling strategy, which has two folds. First, the method assigns datasets with different weights based on its median PPL; then, within each dataset, the method adjusts the percentile based on the loss change of that dataset. Numerical comparisons show that the proposed strategy outperforms other sampling strategies in most scenarios.

**Audience:**

Yes

**Broader Impact Concerns:**

Sampling based on perplexity for LLMs (or other models) may introduce implicit bias and unfairness, which can raise ethical concerns and require further investigation.

**Claims And Evidence:**

No

**Requested Changes:**

In section 4, it missed a space after the word SST.

Please address the weakness above.

**Strengths And Weaknesses:**

Strength:
1. The experiment settings are clear and reasonable. The paper investigates different sampling strategies for models of different sizes and structures.
2. The paper proposed an adaptive sampling strategy, which can adjust the sample selection over a mixture of datasets and within each dataset.

Weakness:
1. On the variance of the runs. The paper reports variance only with two runs for Sec. 3. This is insufficient to show any statistical result.
2. Based on the large variance in Sec. 3, it is hard to justify whether the results in Sec. 4 demonstrate SST's better performance over other cases.
3. Sec. 4.1 is unclear and hard to understand, please consider putting Alg. 1 into this section.
4. It is insufficient to claim aligning data complexity with training progress proves beneficial based on Fig. 2. The paper should provide the comparison between adjusting sampling PPL from high to low, from low to high, fixed low, fixed middle, fixed high. Also, why Llama 3.1-8B misses Keep Top, and Qwen2.5 misses Keep Bottom?
5. Does the method consider the change of sample PPL during training? Or does it use fixed PPL throughout training? If it requires computing $\beta_d$, I presume it requires computing PPL for all data for multiple times. If that is the case, computation in Sec. 4.2 does not make sense to me. The selection cost in eq(4) should be multiplied by the number of adjustments of $\beta_d$.

---

> ### Author Response · Authors · 2025-04-21
>
> ### Weakness 1
> We thank the reviewer for raising the important point regarding variance estimation. We agree that more runs provide stronger statistical grounding.
>
> Using two seeds represents a **deliberate trade-off due to compute availability: prioritizing breadth (diverse architectures and scales) over depth (more seeds per model)**. For context, recent related works such as DEITA (ICLR 2024) evaluated on 2 model sizes (7B, 13B) with 1 seed, and InsTag (ICLR 2024) similarly used 2 13B models with 1 seed.
> The 2-seed results in §3 serve their intended purpose of illustrating the static PPL applicability and limitations across a wide range of models under practical resource limitations.
>
> Recognizing the reviewer's valid concern, we **have increased the seeds from 2 to 4 for the main comparison in Fig. 1**, where SST consistently outperforms these baselines with lower variance, reinforcing the paper's central message.
>
> Adding more seeds would require substantial additional computation (**~1900 GPU/hr/seed, e.g., Fig. 2, requiring ~248 runs)**, **we are prepared to run additional seeds for this section if the reviewer considers it essential** after considering these constraints and the strengthened results in Fig. 1.
>
> ### Weakness 2
> The reviewer has successfully identified the main motivation: the preliminary analysis in (§3) indeed shows considerable variance depending on the configuration, which **aligns with the goal of the section of highlighting static perplexity issues and motivating SST**. Eg, In Fig. 4, the experiment explicitly studies the effect of delaying the sampling start, showing large variance later in training. These results motivates starting SST when the loss stabilizes coinciding with low variance regions.
>
> In response, to robustly evaluate SST, **we rerun the main comparison shown in Fig. 1 using 4 seeds**. As the updated Fig. 1 clearly demonstrates, SST not only achieves consistently higher performance throughout training compared to all baselines but also **exhibits noticeably lower variance**. The baselines, in contrast, often show both lower average performance and higher run-to-run variability.
>
> For clarity, §4 describes the SST method and analyzes its overhead, not SST's main performance.
>
> ### Weakness 3
> We moved Algo. 1 and the hyperparameter table to §4 as suggested.
>
> ### Weakness 4
> We would like to clarify that the comprehensive comparisons for fixed low, middle, and high perplexity segments (static sampling) are provided in Appendix D, specifically in the new Fig. 2 (from the appendix) and 8. Appendix D presents a rigorous ablation across 8 models and 3 subset sizes, evaluating the performance of keeping the bottom (low to high), middle, and top perplexity (high to low) segments following the methodology of Marion et al. both per-dataset and across the entire mixture.
>
> The purpose of Fig. 2 (now Fig 3) is different: it aims to illustrate why an adaptive approach like SST is beneficial, by showing how the relative performance trajectories of different static perplexity segments evolve during the course of training for the two specific models.
>
> **Regarding the missing lines**: for visual clarity in illustrating this dynamic trend, the Fig. intentionally plots only the two best-performing static configurations for each model, as identified from the broader analysis in Appendix D. We **have updated Fig. 2 Caption to explicitly state this**.
>
>
> ### Weakness 5
>
> We agree that the process for updating dataset difficulty during training might not have been perfectly clear initially.
>
> To make this point explicit, we **have added a clarification in the manuscript (page 9)**: "During training, the PPLp50(d) values are efficiently updated based only on examples from the current batch, thus not requiring additional forward passes".
>
> To clarify, SST calculates the per-example PPL values across the entire dataset exactly once, immediately after warm-up. Then βd is recalculated periodically but uses the PPL values derived from the cross-entropy losses from current batches **without the need of additional forward passes**.
> Therefore, the overhead analysis presented in §4.2, accurately reflects the cost structure.
>
> **Re. Space after the word SST:** Thank you for pointing this out. We have corrected the missing space after 'SST'.
>
> **Broader Impact Concerns:**
>
> We sincerely thank the reviewer for raising this crucial point for introducing or amplifying bias. To further address this important aspect, we **added a dedicated Broader Impact statement section to the revised manuscript**.
>
> While acknowledging the general risks, SST, in particular, has the following advantages:
>
> - SST removes risks associated with external model biases, reducing analysis to a single model.
> - SST makes data selection **transparent**, unlike static methods with predefined mix.
>
> A dedicated, large-scale analysis was beyond the scope of the current investigation, which focused on performance and efficiency.

---

### Review · Reviewer_Rpij · 2025-04-07

**Summary Of Contributions:**

1. This paper offers a thorough empirical evaluation of static data selection methods. The analysis is exhaustive and is expected to inspire future work in this area.

2. The proposed method demonstrates strong performance, with performance gains that increase with model size. Additionally, the computational efficiency is commendable.

**Audience:**

Yes

**Claims And Evidence:**

Yes

**Requested Changes:**

1. It is recommended that the authors include some dynamic methods from related works in Table 1.

2. The training dataset is a mixture of many datasets. It would be beneficial to illustrate the ratio of selected data for each sub-dataset for both the proposed method and the baselines.

3. In Table 1, each benchmark includes several subjects, such as math, physics, chemistry, etc. It would be helpful to illustrate the performance differences compared to full data on each subject to better understand the detailed performance gains or losses.

4. Currently, the major benefit of the proposed method is the reduction in training cost. Although there are some performance gains on certain models, they are not significant. In many domains, less data can yield significantly better performance [Ref 1]. It is highly recommended to test the proposed method on this particular dataset.

[Ref 1]  LIMO: Less is More for Reasoning

**Strengths And Weaknesses:**

**Strengths**

1. The discussion of related work is comprehensive and beneficial, helping the general audience quickly understand this field.

2. The experiments on static methods are thorough, and the analysis is insightful. These detailed experiments will support future work in this area.

3. The proposed method demonstrates good performance and computational efficiency.

**Weaknesses**

1. Given the dynamic nature of the proposed method, it would be beneficial to include dynamic method baselines. (Please refer to Change 1)

2. A detailed analysis of Tables 1 and 2 would be insightful for readers. (Please refer to Changes 2, 3)

3. The impact of the paper could be enhanced by applying it to domains where less data is more advantageous. (Please refer to Change 4)

---

> ### Author Response · Authors · 2025-04-21
>
> **Change Request 1**:
> We thank the reviewer for this valuable suggestion.
> In response, we have **added RHO-Loss**, a prominent adaptive baseline from recent literature, to our experimental comparisons. Specifically, the updated Table 2 on Page 12 (previously Table 1) now includes performance results for RHO-Loss alongside other baselines and our proposed SST method. Additionally, Figure 1 visualizes performance trends throughout the training process, clearly illustrating how RHO-Loss compares to SST and other methods. SST remains the best-performing method overall. Additionally, we have conducted experiments with two additional random seeds per method to better characterize variance and ensure robustness of our findings.
>
> Further, in Section 5.2, we added a detailed analysis about the comparative behavior and performance characteristics of RHO-Loss relative to SST, highlighting the differences and operational implications.
> We appreciate this insightful recommendation, which we believe has substantially enhanced the depth and rigor of our experimental evaluation.
>
> **Change Request 2:** We thank the reviewer for this insightful suggestion, as it enriched the interpretability and depth of our experimental results.
> we have **added Figure 7 to the manuscript**. This figure explicitly visualizes the final percentage distribution of selected sub-datasets for all static methods. Additionally, for dynamic methods (RHO-Loss${30k}$ and our SST$_{30k}$), Fig 7 illustrates how the composition of the selected data evolves throughout the training process at different iterations.
> This visualization clearly highlights operational differences and selection biases among the methods, reinforcing our analysis in Section 5.2. Notably, SST (Fig 7c) appears to maintain a broader representation of datasets throughout training. This sustained diversity likely contributes to SST’s more robust and consistently superior performance.
>
> **Change Request 3:** For Table 1 (now Table 2), we followed the standard reporting format of the Open LLM Leaderboard, presenting the established aggregate scores. This ensures consistency and allows for direct comparison with standard leaderboards.
> While we agree that a subject-level breakdown could offer more detailed insights, the current evaluation harness and leaderboard reporting provide results at a very granular level (e.g., IFEval reports instruction-level accuracy, BBH comprises 23 sub-tasks, MMLU-PRO reports only an overall accuracy, etc.) (eg, https://huggingface.co/datasets/open-llm-leaderboard/MaziyarPanahi__calme-3.2-instruct-78b-details). Incorporating dozens of these fine-grained sub-scores (**36 in total**) directly into Table 2 would would significantly increase its complexity and reduce readability, potentially obscuring the key insights and primary comparisons.
>
> Nevertheless, we fully acknowledge the importance of this detailed analysis. If the reviewer still finds this granularity essential, we would be happy to include the detailed per-subject or per-task performance breakdown as an appendix in the final version of the manuscript. We believe this approach strikes a suitable balance between clarity in our primary presentation and the availability of comprehensive results for readers interested in deeper insights.
>
> **Change Request 4:** We thank the reviewer for their valuable feedback and the suggestion to evaluate our proposed method on specialized reasoning datasets, such as the one referenced (LIMO).
>
> We fully recognize the potential value of exploring how our method performs specifically in domains where less data can significantly enhance reasoning capabilities. However, our current study focuses explicitly on general instruction fine-tuning across diverse tasks and models, utilizing the comprehensive Open LLM Leaderboard v2 benchmark suite. Expanding our evaluation to specialized reasoning benchmarks would necessitate additional substantial experimental efforts (potentially incorporating ~10 new specific tasks like MATH500, AIME)  and computational resources beyond our current scope, especially considering that we prioritized addressing the reviewer’s earlier comment by incorporating an adaptive method (RHO-Loss), which consumed significant computational resources during the rebuttal period. We hope the reviewer understands these constraints and agrees that our current evaluation provides substantial evidence for SST's effectiveness in the general IFT setting.
>
> While we highlight the efficiency gains and training cost reduction as significant benefits of SST, we also respectfully point out that the performance gains, particularly on larger models, are consistent and often substantial compared to strong baselines using the same data budget. For instance, as shown in Table 2, SST$_{30k}$ achieves a +4.54% average score improvement over the next best 30k baseline on Qwen2.5-32B and outperforms the baseline trained on the full 100k dataset on 5 out of 7 models tested.

---

### Author Response · Authors · 2025-04-21

Dear Reviewers,

We sincerely thank you for dedicating your time and expertise to reviewing our manuscript. Your insightful comments and constructive feedback have been invaluable in helping us improve the quality and clarity of our work.

We wanted to briefly note that, as part of our revisions to address your feedback and enhance the paper, we have incorporated one additional figure and one additional table into the manuscript. As a result, **please be aware that the numbering for figures and tables subsequent to these additions may differ from their numbering in the originally submitted version**.

We believe the changes made have significantly strengthened the paper, and we are grateful for your guidance throughout this process.

---

### Decision · Action_Editor_9NEh · 2025-05-16

**Recommendation:** Accept as is

**Comment:**

This paper proposes a technique for data selection in large language models that makes use of the per-example perplexity assigned by the model itself. Overall, the idea is straightforward, and the results are convincing - it outperforms various data selection baselines across various settings. The reviewers all recommended acceptance. My only suggestion would be to better clarify (in the title or introduction or wherever) that you primarily focus on data selection for instruction tuning in this work.

**Audience:**

Yes.

**Claims And Evidence:**

Reviewers all found the claims to be convincing and the experimental setting sound. One reviewer asked for more theoretical justification and was satisfied by the rebuttal update.

---

> ### Author Response · Authors · 2025-06-08
>
> Thank you for your positive feedback and helpful suggestion. In response to your comment, we revised the text to clearly indicate that the work focuses on data selection for instruction fine-tuning (IFT), both in the abstract ("Extensive experiments on seven LLMs (0.5B to 32B parameters) **in the instruction-finetuning (IFT)** setting show that Sst ...") and in the contribution section ("We demonstrate Sst’s effectiveness in the IFT setting on seven models ..."). We chose not to update the title, as the method is also applicable to pre-training. However, we believe the revisions make the evaluation context clear to the reader.